# Discovery and mechanism-guided engineering of BHET hydrolases for improved PET recycling and upcycling

Anni Li[1,4], Yijie Sheng[1,4], Haiyang Cui[2,3,4], Minghui Wang[1], Luxuan Wu[1], Yibo Song[1], Rongrong Yang[1], Xiujuan Li ®[1] ✉ & He Huang ®[1] ✉

Although considerable research achievements have been made to address the plastic crisis using enzymes, their applications are limited due to incomplete degradation and low efficiency. Herein, we report the identification and subsequent engineering of BHETases, which have the potential to improve the efficiency of PET recycling and upcycling. Two BHETases (ChryBHETase and BsEst) are identified from the environment via enzyme mining. Subsequently, mechanism-guided barrier engineering is employed to yield two robust and thermostable ΔBHETases with up to 3.5-fold enhanced $k_{cat}/K_M$ than wild-type, followed by atomic resolution understanding. Coupling ΔBHETase into a two-enzyme system overcomes the challenge of heterogeneous product formation and results in up to 7.0-fold improved TPA production than seven state-of-the-art PET hydrolases, under the conditions used here. Finally, we employ a ΔBHETase-joined tandem chemical-enzymatic approach to valorize 21 commercial post-consumed plastics into virgin PET and an example chemical (*p*-phthaloyl chloride) for achieving the closed-loop PET recycling and open-loop PET upcycling.

Plastics have facilitated human life and become an integral part of packaging, healthcare, aerospace, and agriculture due to their durability, transparency, and economic viability[1]. Due to the lack of or low activity of catabolic microorganisms/enzymes that can degrade these plastic constituents, many plastics persist significantly in terrestrial and marine environments[2-5]. Microplastics have even been found in human feces, blood, and placenta, indicating that plastic pollution is a threat to human health and needs more attention[2,3,6-8]. Polyethylene terephthalate (PET), a significant portion of plastic waste[9-11], is exceedingly hard to degrade due to the chemical inertness of the ester linkages and the aromatic nuclei[4,12,13].

Compared to traditional physical and chemical processes[14-16], biodegradation using microbes and enzymes provides a greener and more sustainable approach with less energy consumption, lower $CO_2$ emissions, and no groundwater contamination[14]. Although several PET-degrading microorganisms (e.g., *Ideonella sakaiensis*[9], *Thermobifida cellulosilytica*[17], *Saccharomonospora viridis*[18], *Fusarium solanipisi*[19], *Nocardiopsaceae*[20]) and enzymes (e.g., lipase[19,21], esterase[22-25], cutinase[19,26-28], PETase[9,29], and mono-2-hydroxyethyl terephthalate hydrolase (MHETase)) have been well explored[9,30], researchers are still mining for the last piece of the puzzle, bis-2-hydroxyethyl terephthalate hydrolase (BHETase), to complete the PET biodegradation landscape[31] (Fig. 1a). Current PET recycling without BHETase and/or relevant bacteria still faces the bottleneck that the heterogeneous products yielded from PET degradation are unfavorable for PET recondensation and high-value derivative synthesis. Since the scope for chain extension during polymerization production is restricted to different end groups[32,33], the mechanical properties of

[1]School of Food Science and Pharmaceutical Engineering, Nanjing Normal University, Nanjing 210009, People's Republic of China. [2]RWTH Aachen University, Templergraben 55, Aachen 52062, Germany. [3]Present address: University of Illinois at Urbana-Champaign, Carl R. Woese Institute for Genomic Biology, 1206 West Gregory Drive, Urbana, IL 61801, USA. [4]These authors contributed equally: Anni Li, Yijie Sheng, Haiyang Cui. ✉e-mail: lixiujuan@njnu.edu.cn; huangh@njnu.edu.cn

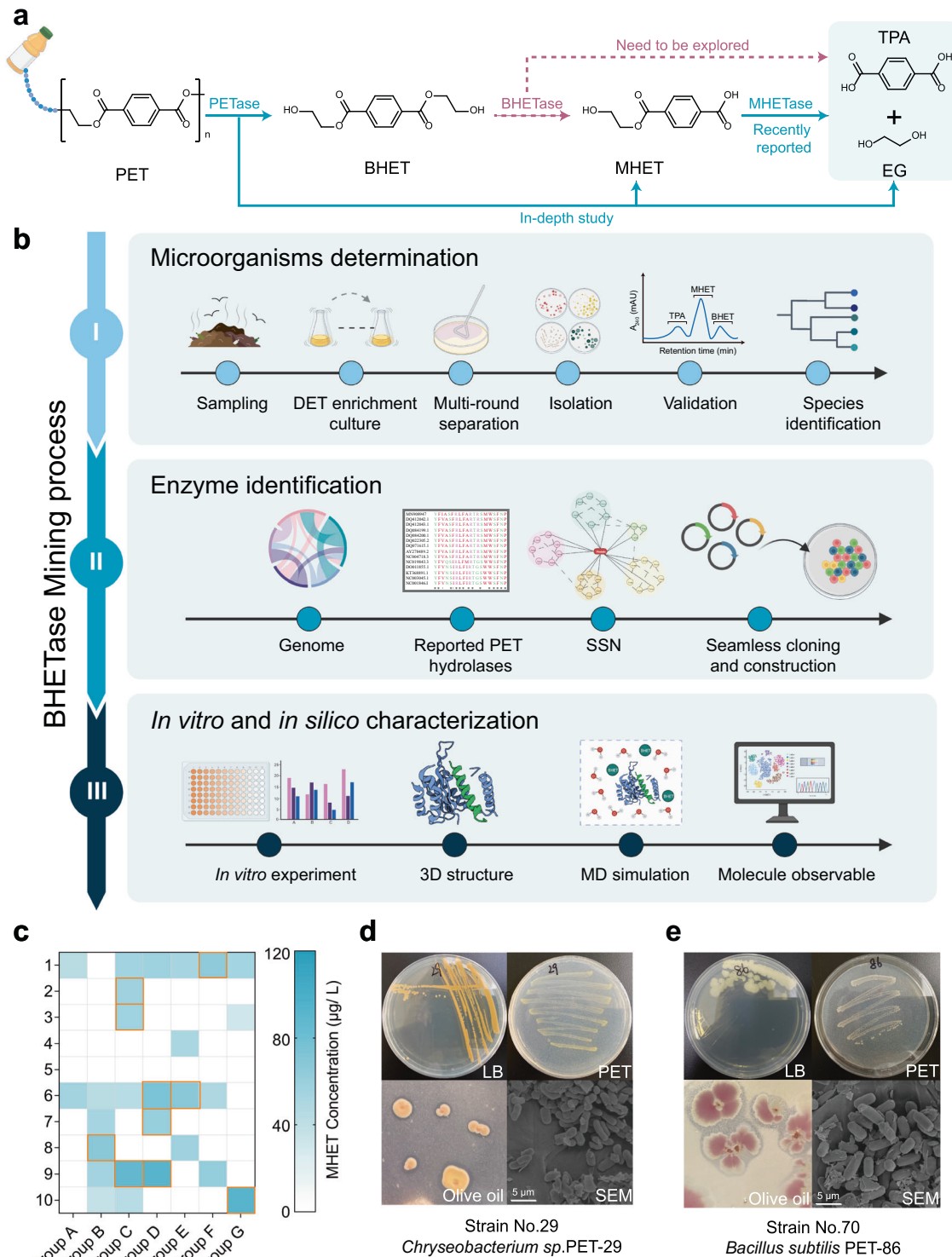

**Fig. 1 | The enzyme mining process for identifying BHETase. a** Except for BHETase, both PETase, and MHETase have been well studied. PET: polyethylene terephthalate; TPA: terephthalic acid; EG: ethylene glycol; BHET: bis-2-hydroxyethyl terephthalate; MHET: mono-2-hydroxyethyl terephthalate. **b** An overview of the BHETase mining process to screen and identify microbes and enzymes with PET degradability. Icon graphics of this figure was created by BioRender.com. **c** The obtained 70 microbes with degrading PET film after multi-round separation and isolation. Reaction conditions: pH 7.5, 30 °C, 2 g/L PET powder, and 5% inoculum for 5 days. All measurements were conducted in triplicate ($n = 3$), and the mean values were used for generating the heat maps. **d** The overview of *Chryseobacterium sp.* PET-29 (No. 29) and **e** *Bacillus subtilis* PET-86 (No. 70) were cultivated in LB agar plates, salt agar plates with PET powder, and olive oil plates, respectively. The scanning electron microscopy (SEM) images of *Bacillus subtilis* PET-86 (No. 70) and *Chryseobacterium sp.* PET-29 (No. 29) cells were grown on PET powder. Results were reproduced three times independently; representative micrographs are shown.

recycled PET made with heterogeneous polymers (e.g., the mixture of BETH, MHET, and TPA) are often not ideal[34]. Once discovered, microorganisms and enzymatic machinery able to break down BHET could be employed as a base for fermentation and/or for the entire biological recycling of PET waste.

Several protein engineering efforts have been implemented to further improve the catalytic properties of wild-type PET hydrolases[35–40]. For instance, the evolved PETases with impressive thermostability (ThermoPETase[40], DuraPETase[39], and FAST-PETase[37]) were obtained through directed evolution, (semi-)rational design, and machine learning approaches. Also, the engineered LCC-ICCG cutinase showed a minimum of 90% PET depolymerization into monomers over 10 h by introducing the disulfide bridge[35]. Interestingly, MHETase can be endowed with BHETase activity by modifying the distal part of the binding pocket, but such enzymes still exhibit low activity[41,42]. Unfortunately, the drawback of the heterogeneous product cannot be solved by adding extensive PETases, because PET hydrolases often possess large hydrophobic surfaces to assist themselves in binding the PET surface, which results in dramatically reduced free enzymes and incomplete degradation of dissolved BHET and MHET. Recently, mechanism-guided enzyme engineering, benefiting from the power of combining computational dynamics studies with directed evolution, showed remarkable success in studying the sequence-structure-function relationship[43], which provides a complementary method for enhancing PET biodegradation performance by overcoming the challenge of incomplete degradation.

Herein, we identify two BHETases by using a three-stage enzyme mining process: (i) microorganism determination; (ii) enzyme identification; (iii) in vitro and in silico characterization. The knowledge obtained in the third step encouraged us to apply mechanism-guided barrier engineering to rationally tailor the obtained BHETases (BsEst and ChryBHETase) for improving the catalytic properties and to yield a pure product, TPA. Furthermore, we utilize the engineered ΔBHETases (ΔBsEst and ΔChryBHETase) in a two-enzyme system to generate homogeneous TPA through PET biodegradation. We demonstrate the feasibility of coupling ΔBHETases with all seven state-of-the-art PET hydrolases, namely PETase[9], DepoPETase[44], FAST-PETase[37], DuraPETase[39], ThermoPETase[40], LCC[28], and LCC-ICCG[35]. As biotechnological degradation and recycling of plastics are still in the early stages of the learning curve, we further develop a tandem chemical and biological PET degradation system that leverages the advantages of both depolymerization processes to expand the applicability of BHETases. To achieve fully closed-loop PET recycling and open-loop PET upcycling, respectively, the obtained pure TPA is used for resynthesizing recycled PET and valorizing TPA into a valuable chemical product (p-phthaloyl chloride as an exemplary chemical).

## Results

### Two BHETases were identified from the environment through enzyme mining

Conventional enzyme mining pipelines can be laborious and off-target in identifying PET hydrolases[45]. In this section, we report an enzyme mining process which includes three phases (Fig. 1b), which was applied to identify PET hydrolases.

In Phase I (microorganism determination) and Phase II (enzyme identification), we collected 50 samples from refuse landfills, including PET debris and the surrounding soil. The common substrate diethyl terephthalate (DET) is used for enriching PET degradation microorganisms, because DET is a small molecule structurally similar to PET[46]. DET can be easily hydrolyzed by PET degradation microorganisms, releasing terephthalic acid and ethanol, which can be detected by various analytical methods like high-performance liquid chromatography (HPLC)[47–49]. By using the diethyl phthalate (DET) and PET powder for stepwise strain isolation followed by MHET detection (Fig. 1c, Supplementary Figs. 1–3 and Supplementary

Table 1), two promising strains *Chryseobacterium sp.* PET-29 (GenBank ID: OP564169, Fig. 1d) and *Bacillus subtilis* PET-86 (GenBank ID: OP564167, Fig. 1e), were identified from 70 microbes with up to 162.58 μg/L MHET yields in 48 h. In contrast to *Bacillus sp.*, which was found in the previous study[25], *Chryseobacterium sp.* PET-29 was identified here to exhibit PET degradation. Genome sequencing was subsequently performed to assist the PET hydrolase mining (Fig. 2a, Supplementary Fig. 4, and Supplementary Tables 2 and 3). In the next phase, helped by sequence similarity networks (SSNs) analysis, we clustered the 85 open reading frames (ORFs) labeled with α/β-hydrolase domain in both strains with 29 reported PET hydrolases (Fig. 2b and Supplementary Tables 4 and 5) and successfully identified two enzymes, BsEst (Supplementary enzyme sequence 1, 2) and ChryBHETase (Supplementary enzyme sequence 3, 4). More details about Phase I and II are described in Supplementary Notes 1 and 2.

In Phase III, sequence identity analysis and in vitro characterization of the purified BsEst and ChryBHETase with degrading PET powder, BHET, and pNP-aliphatic esters confirmed that ChryBHETase prefers BHET to aliphatic esters (Fig. 1d, e, Fig. 2c–e, and Supplementary Figs. 5–7), compared with lipase (BSLA, identity of ChryBHETase vs. BSLA is 13.18%,), esterase (Bs2Est, identity 25.55%; TfCa, identity 24.66%,), PETase (identity 12.41%) and MHETase (identity 14.31%)[9,30] (Fig. 2c and Supplementary Figs. 8–10), leading to its designation as the BHET hydrolase (termed BHETases). BsEst belongs to one kind of esterase regarding the sequence identify analysis (Supplementary Fig. 9c), but its preferred BHETase activity encourages us to treat it as a BHETase. The latter phenomenon was also found by a recent study[50] that a promiscuous carboxylesterase called TfCa can hydrolyze BHET and MHET (identity of TfCa vs. BsEst is 37.21%, Fig. 2c, and Supplementary Fig. 10).

The computational study, especially molecular dynamics (MD) simulation, can provide a molecular understanding of the identified enzymes[43,51–55]. Towards in silico characterization of Phase III, the predicted 3D structure of ChryBHETase by AlphaFold2 lacked two regions (residues 251–272 and 245–369) compared to BsEst and TfCa, as shown in sequence alignment (Fig. 2c and Supplementary Fig. 10). Besides, the electrostatic potential energy distribution analysis of ChryBHETase, especially at the substrate binding cleft (SBC, Fig. 2g), showed a completely different pattern compared to esterase BsEst, emphasizing the specific feature of ChryBHETase when working on BHET. Besides, we performed the MD simulation of BsEst and ChryBHETase in the absence/presence of substrate BHET to investigate the PET hydrolase-substrate interaction (Supplementary Figs. 11 and 12). After examining 20 structural and solvation observables (Supplementary Table 10, Fig. 2h, i, and Supplementary Figs. 13–21, see Supplementary Note 7), we observed that the overall structures of BsEst and ChryBHETase are stable in water (Fig. 2h and Supplementary Figs. 13–17). Almost no BHET molecule could be detected in the SBC of BsEst and ChryBHETase (Fig. 2i). However, a certain number of water molecules were present in SBC, indicating the larger BHET might be blocked by uncertain barrier structures (the particular region is called lid in hydrolase[56–58]). These results were confirmed by the spatial distribution function (SDF) of BHET and water around the protein surface (Fig. 3b).

### Mechanism-guided barrier engineering yielded robust BHETase

The native catalytic efficiency and stability of wild-type enzymes often limit their wide application in industry[59]. Regarding the solvation phenomena around the SBC, we hypothesized that removing the barrier may favor BHET access and increase degradation efficiency, followed by proposing mechanism-guided barrier engineering (Fig. 3a). The identified barrier regions in each BsEst (Δ1: residue 60–77, Δ2: 105–112, Δ3: 267–275, Δ4: 401–418, and Δ5: 410–418, Supplementary Fig. 22a) and ChryBHETase (Δ1: 63–78, Δ2: 66–76, Δ3: 267–274, Δ4: 319–327, and Δ5: 368–376, Supplementary Fig 23a) were

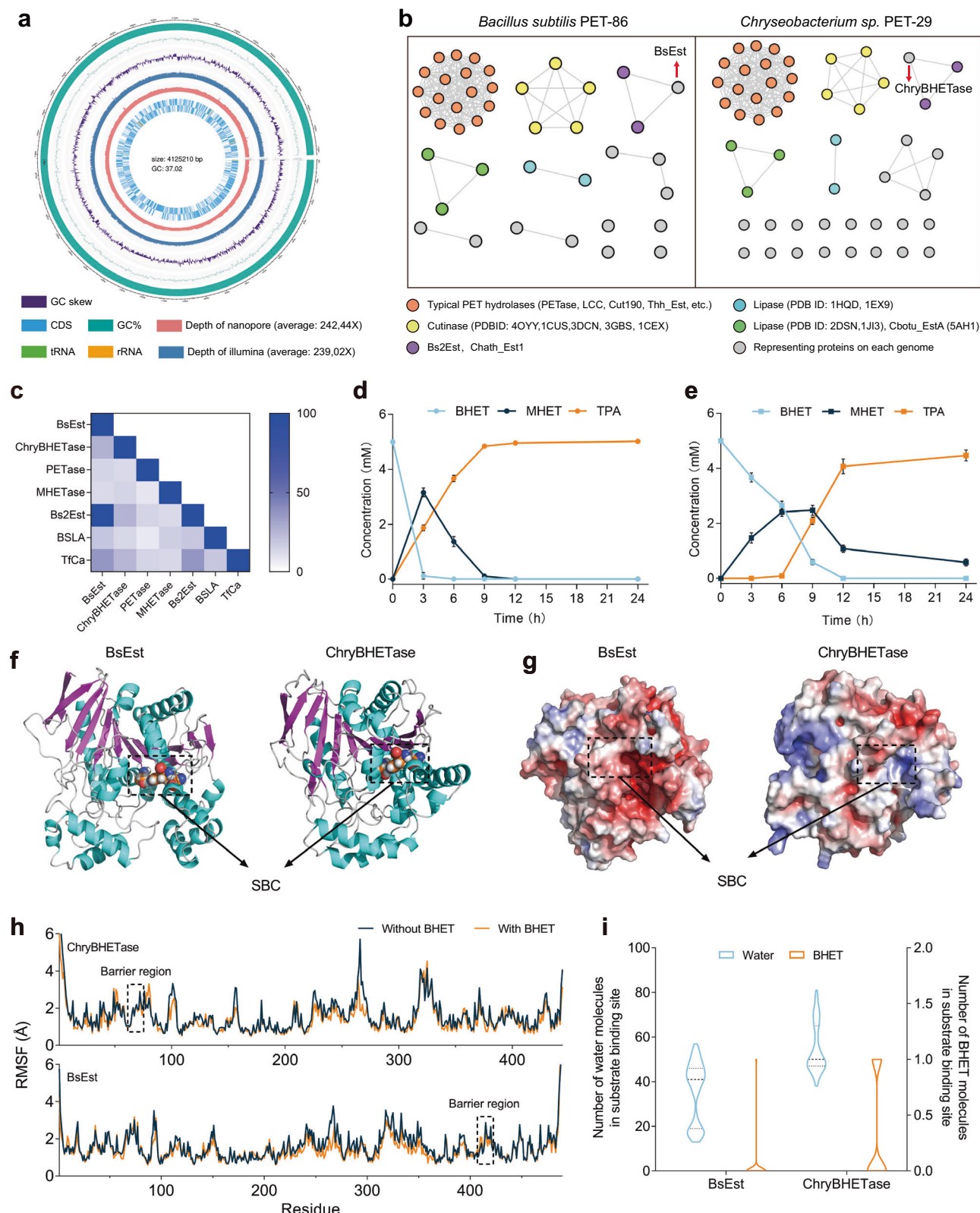

decided to be truncated and replaced by a flexible linker (-GG-). Then, in silico scoring criteria were performed by considering five factors in substrate docking studies, including binding energy, number of hydrogen bonds, serine affinity attack distance, and cavity volume (Fig. 3c and Supplementary Figs. 22–24). The top variant BsEst-Δ5 (named as ΔBsEst, Fig. 4a) and ΔChryBHETase-Δ2 (named as ΔChryB-HETase, Fig. 4b) were validated by in vitro experiments.

As expected, ΔBsEst and ΔChryBHETase showed greater catalysis efficiency than wild-type (Fig. 3d–f). ΔBsEst completed 100% BHET conversion in 3 h, which is a 2-fold improvement over BsEst (6h, Fig. 3d). And ΔChryBHETase achieved 100% conversion of BHET 3h earlier than ChryBHETase. ΔBsEst converted almost all intermediate MHET to TPA within 6 h, whereas ~50% of MHET was still resting via BsEst. Similar results were observed for ΔChryBHETase. Besides,

**Fig. 2 | Enzyme identification through sequence similarity networks (SSN) and enzyme determination by in vitro biodegradation performance and in silico computational examination. a** Circular genomic map of *Chryseobacterium sp.* PET-29. Functional proteins were determined by the BLAST software against the COG database, and enrichment analysis of molecular function, cellular components, and biological process was identified using the GO database. **b** The sequence similarity networks (SSN) analysis of PET hydrolase in *Bacillus subtilis* PET-86 and *Chryseobacterium sp*. PET-29. Each protein is represented by a node. The proteins within the same functional family are connected by the edge to form a cluster. The nodes representing BsEst and ChryBHETase are labeled with an arrow. **c** Sequence identity results of BsEst and ChryBHETase with other reported enzymes. Biodegradation performance of **d** BsEst and **e** ChryBHETase upon incubation on BHET. Reaction condition: 5 mM BHET at 30 °C in pH 7.5 buffer for 24 h. Error bars

correspond to the standard deviation (s.d.) of three measurements ($n = 3$). **f** 3D structures of BsEst (left) and ChryBHETase (right) predicted by AlphaFold2. The sphere model represents the catalytic triad. S189-E310-H399 of BsEst; S194-E310-H379 of ChryBHETase. The black dashed line represents the substrate binding cleft (SBC). **g** The surface electrostatic potential distribution of BsEst (left) and ChryB-HETase (right). The positive charge is shown in blue, and the negative charge is shown in red. **h** RMSF of BsEst and ChryBHETase residues determined from the last 40 ns of MD simulation in water and BHET systems. Data plotted from the average of three independent MD runs ($n = 3$). **i** The number of BHET/water molecules at the substrate binding site. The dotted line represents the quartiles of the data, and the middle-dotted line represents the median. Data plotted from the last 40 ns of three independent MD runs ($n = 243$).

ΔChryBHETase can produce TPA immediately at the beginning of BHET degradation and achieve > 90% yield within 12 h. In contrast, ChryBHETase had just begun to accumulate TPA after 6 h. Eventually, both ΔChryBHETase and ΔBsEst produced 100% TPA more efficiently than untruncated ones. A similar observation was also confirmed by specific activity and kinetic study (Table 1). Both ΔChryBHETase and ΔBsEst showed significantly increased turnover numbers ($k_{cat}$, ΔChryBHETase: 2.6-fold, ΔBsEst: 1.7-fold) compared to ChryBHETase and BsEst. The catalytic efficiency ($k_{cat}/K_M$) of ΔBsEst and ΔChryBHETase had 3.5-fold and 1.5-fold enhancement, respectively. Additionally, the thermostabilities of ΔBsEst and ΔChryBHETase were able to retain over 80% of initial activities after 2 h of incubation at 30–50 °C, which is better than the truncated enzymes under the same conditions (Supplementary Fig. 24). As indicated in Supplementary Table 9, both BHETase and ΔBHETase maintained substantial activity within the temperature range of 50 °C to 60 °C. Notably, the engineered ΔBHE-Tase variant ΔBsEst demonstrated superior thermostability than the wild type, with a half-life of up to 38.23 h at 60 °C. These findings underscore the potential of ΔBHETase to sustain activity and stability even at elevated temperatures, positioning it as a candidate for PET degradation applications. These results suggested ΔBsEst and ΔChryBHETase do not destroy the stability of structure and hold the potential to be applied in practical biodegradation (see Sections 2.4 and 2.5). Also, such kinetics and thermostability improvements should benefit from the removed barrier, as confirmed by the subsequent MD simulation studies (Section 2.3).

### Removing the barrier empowered the ΔBHETases (ΔBsEst and ΔChryBHETase)

To demonstrate the driving force behind the increased degradation capacity, we also investigated the MD simulation of ΔBHETase (ΔBsEst and ΔChryBHETase) in the presence of BHET (Supplementary Figs. 11 and 12). We found that the truncated barrier in BsEst and ChryBHETase has low flexibility (Fig. 2h, Supplementary Fig. 21, and 25 RMSF: 1-3 Å), which is different from the flexible lid in lipase that distinguishes barrier engineering from lid engineering[57,58,60,61]. The rigid barrier is not favorable for the access of substrate into SBC. In addition, we did find that the BHET appeared near the barrier in BsEst and ChryBHETase (Supplementary Fig. 18). After truncation, BHET density appeared in the SBC, as shown in the SDF study (Fig. 3b), suggesting BHET can enter the SBC easier than in BsEst and ChryBHETase.

To investigate the dynamic binding process of BHET molecules into SBC, we calculated the distance change between nucleophilic Ser in the catalytic triad and substrate BHET as a function of time (100 ns, Fig. 4c–f, Supplementary Fig. 26). Compared to the BHETase-BHET system, BHET molecules in both ΔBsEst- and ΔChryBHETase-BHET systems kept getting closer to SBC as time increased (Fig. 4c–f). One BHET molecule existed in the SBC after 40 ns in both ΔBHETase-BHET systems (Fig. 4g, h). Interestingly, the number of BHET molecules in SBC of BHETases (especially ChryBHETase) slightly increased during 60–100 ns, yet the occurrence probability of BHET in SBC became

more significant after removing the barrier (around 2.5–6.6 fold improvement in the last 10ns), agreeing well with the kinetic study ($k_{cat}/K_M$, Table 1). These molecular observations confirmed the improved catalytic performance of ΔBsEst and ΔChryBHETase mainly results from the removed barrier region.

Besides, the overall and local structural changes of two ΔBHE-Tases were further studied by investigating several observables (e.g., RMSD, $R_g$, internal H-bond, SASA, RMSF, Supplementary Figs. 13–17, and 19–21). The results demonstrated that ΔBsEst and ΔChryBHETase become slightly more stable than BHETases with the barrier (decreased RMSD value in Supplementary Figs. 13–17), which explains the moderately enhanced thermostability (Supplementary Fig. 18). Besides, the finetuned conformation of SBC in both ΔBHETases might be favorable for releasing products, which was revealed by the docking studies (Supplementary Figs. 19, 20 and 27, 28; see Supplementary Note 8).

### Full TPA conversion from PET achieved by ΔBHETase in a simple two-enzyme degradation system

In 2016, Yoshida et al. published a groundbreaking study introducing *Is*PETase[9]. This enzyme exhibited superior activity to TfH, LCC, and FsC when degrading commercial bottle-derived PET and PET film under ambient conditions. This discovery shed light on the potential to mitigate the environmental impact of uncollectable PET release, offering a promising avenue beyond current recycling methods[62–65]. Nevertheless, TPA produced by PETase often suffered from contamination by oligoethylene terephthalates, BHET, and MHET, which posed limitations on downstream applications[66–68]. To address this challenge, we have developed a two-enzyme degradation system that combines different PET hydrolases, including PETase[9], DepoPETase[44], FAST-PETase[37], DuraPETase[39], ThermoPETase[40], LCC[28], and LCC-ICCG[35], with ΔBHETase. This system enables the production of pure TPA, as illustrated in Fig. 5a.

To investigate the viability of ΔBHETase at high temperatures, we conducted experiments under a representative reaction condition of 60 °C. This choice of temperature aligns with commonly employed conditions in various PET degradation studies[35–37,44]. As depicted in Fig. 5b–h, coupling ΔBHETase with all seven PET hydrolases in a two-enzyme system resulted in a substantial increase in TPA production compared to using PET hydrolase alone in a single-enzyme system. Interestingly, our study did not observe complete hydrolysis of PET films into homogeneous TPA within 120 h for any single PET hydrolase under the current depolymerization conditions (as shown in Fig. 5b–h). Instead, significant amounts of BHET and MHET were detected during degradation in almost all single-enzyme systems at 96 h (Fig. 5i). Similar results were observed by Jiang et al. when using LCC-ICCG[69], but slightly differ from previous studies on single FAST-PETase[37], and LCC-ICCG[35]. The enzymatic reaction conditions (e.g., enzyme amount and purity, buffer type, pH) and the physical-chemical properties of specific PET (e.g., crystallization, absorption, scalping, intrinsic viscosity[70–73]) are not identical between these studies and such factors may be contributing

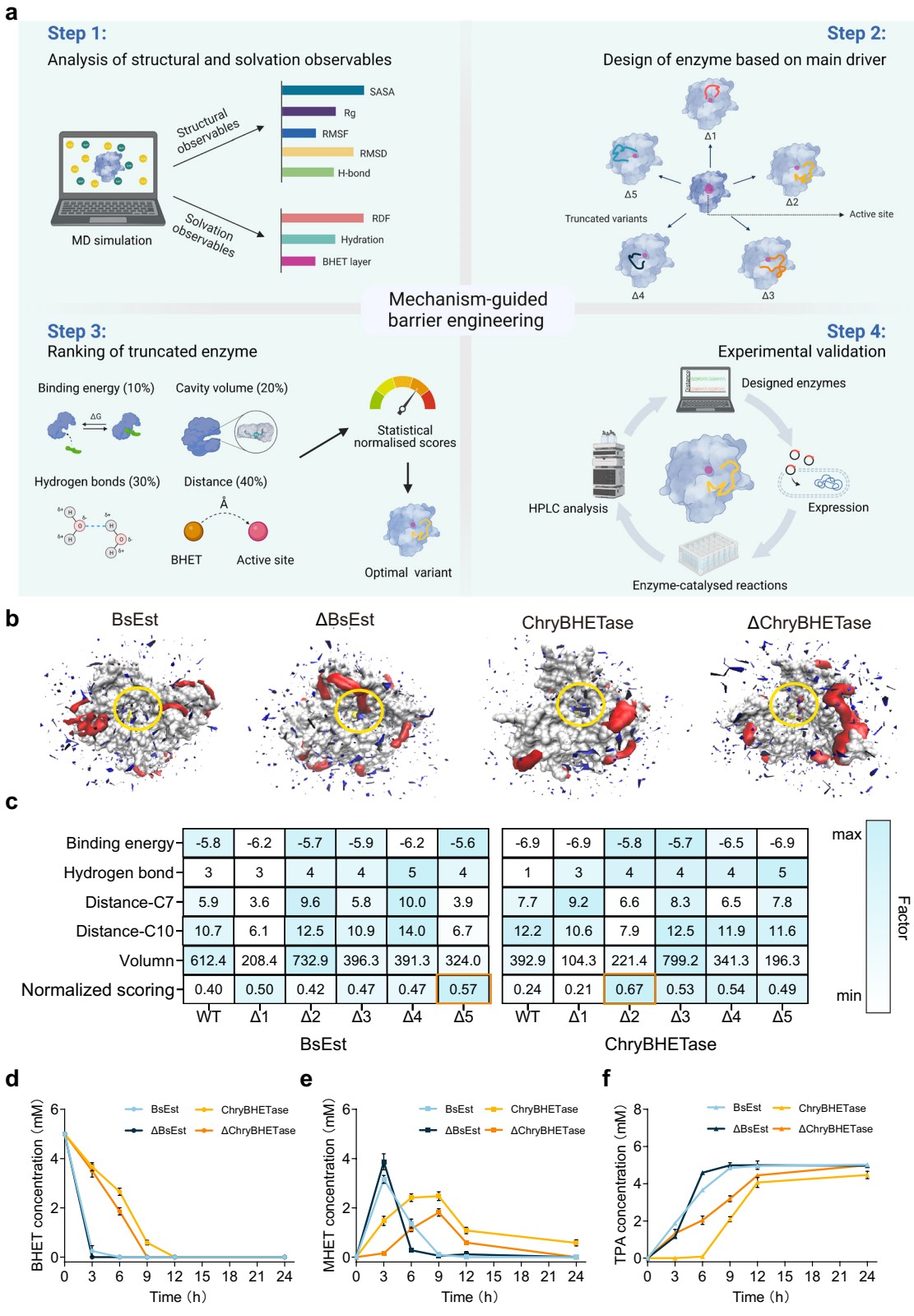

factors to the different outcomes. In contrast, the incorporation of ΔBHETases into a single depolymerization system at 60 °C resulted in the complete hydrolysis of PET films to TPA, as observed in DepoPETase-ΔBHETase and FAST-PETase-ΔBHETase (Fig. 5c, d and Supplementary Fig. 31). Specifically, when combined with DepoPETase, ΔBsEst demonstrated a 1.8-fold TPA yield increase compared to Depo-PETase alone. Similarly, in combination with FAST-PETase,

ΔChryBHETase exhibited a 1.6-fold increase, and ΔBsEst achieved a 2.0-fold increase compared to FAST-PETase alone. These notable enhancements can be attributed to the ΔBHETases higher affinity for the intermediate product BHET and their ability to maintain a superior turnover efficiency compared to the state-of-the-art PET hydrolases (Supplementary Fig. 32, TPA conversion rate of ΔBsEst is 31.6 mM/h/mg_enzyme, and ΔChryBHETase is 14.4 mM/h/mg_enzyme). This discovery

**Fig. 3 | The mechanism-guided barrier engineering yielded the improved ΔBHETase. a** Schematic representation of the mechanism-guided barrier engineering strategy. Step 1: In silico prediction of structure and solvation observables. Step 2: Design truncated variants. Step 3: Statistical normative scores by assigning weights of 10%, 30%, 40%, and 20% to the binding energy, number of hydrogen bonds, serine affinity attack distance, and cavity volume, respectively. Step 4: In vitro validation. Icon graphics of this figure was created by BioRender.com. **b** Spatial distribution of BHET and water occupancy at the surface of the optimal variants (ΔBsEst and ΔChryBHETase) and their wide type in BHET simulation systems. Grey: enzyme surface; yellow: serine (the catalytic triad); red: BHET molecule with a solid surface; blue: water molecules with a solid surface. **c** The normalized scorings of truncated BsEst variants and ChryBHETase variants through molecular docking. The binding energy of different receptor proteins to the ligand (BHET) is

calculated by molecular docking. The number of BHET molecules generating hydrogen bonds with surrounding amino acid residues. The distance between the oxygen of the Ser hydroxyl group and the carbonyl carbon (C7 and C10) of the BHET molecule, respectively. The cavity volume formed between BHET and the amino acid residues within 4 Å that was calculated by parKVFinder. Since the factors played different priorities, the normalized scoring was obtained by assigning weights of 10%, 30%, 40%, and 20% to the binding energy, the number of hydrogen bonds, distance, and volume, respectively. Comparison of the activities of ΔBsEst and ΔChryBHETase upon **d** BHET consumption, **e** intermediates MHET generation, and **f** terminal products TPA generation. Reaction condition: 5 mM BHET at 30 °C in pH 7.5 buffer for 24 h. Error bars correspond to the standard deviation (s.d.) of three measurements ($n = 3$).

underscores the capability of ΔBHETase to serve as a pivotal catalyst in the PET degradation process.

The discovery mentioned above was further supported by the analysis of the PET film surface using visible photography, scanning electron microscopy (SEM), and the measurement of reduced the water contact angle (Fig. 5j). Specifically, the two-enzyme system exhibited a remarkable ability to effectively degrade the PET film with up to a 22° water contact angle decrease, resulting in a visibly rougher surface compared to the single enzyme system. To address the additional energy costs associated with high temperatures, we also investigated the PET degradation efficiency of the two-enzyme system at room temperature (30 °C) to achieve a more environmentally friendly process. And we observed similar achievements at room temperature compared to 60 °C (Supplementary Fig. 33). Notably, PETase/ΔBsEst and PETase/ΔChryBHETase produced 663.1 µM (7.0-fold) and 617.6 µM (6.5-fold) of TPA within 24 h, far more than PETase alone (95 µM). Summarily, these achievements reinforce our conviction that the discovered BHETase could play a crucial role in the puzzle of PET biodegradation patterns. Further details regarding the two-enzyme system can be found in Supplementary Note 9 and Supplementary Fig. 33.

### The closed-loop recycling and open-loop PET upcycling based on the BHETase-joined tandem chemical and biological PET degradation system

Although our results and similar works confirmed that biotechnological degradation has potential for PET recycling[74,75], enzymatic hydrolysis often has low substrate loads and slower PET degradation compared to certain thermochemical methods[42,75–77]. To fully explore the potential of applying ΔBsEst and ΔChryBHETase into practical PET re/upcycling, we developed a tandem chemical-biological approach to leverage the advantages of chemical and biological depolymerization processes. This approach involves a chemical pretreatment method called glycolysis at the first stage to efficiently degrade PET into low molecular weight hydrolysate (BHET) within a short time frame while minimizing excessive decomposition[78,79]. Subsequently, ΔBHETases contributed to yielding pure TPA from the intermediate BHET.

As shown in Fig. 6a, potassium carbonate ($K_2CO_3$) as a chemical catalyst was selected due to its biocompatibility compared to the most active, but toxic, glycolysis catalysts, Zn salts[80]. Twenty-one post-consumer plastic products were collected for practical examination, including beverage packaging, food packaging, and household plastic fields (Supplementary Table 11). To evaluate our tandem degradation strategy, a large sample pool of 21 commercial post-consumed plastic wastes with crystallinities ranging from 1.54% to 38.73% was used (Supplementary Table 11). Significantly, the BHET yield obtained from PET glycolysis reached a level of over 81.5% (Supplementary Figs. 34a and 35). Moreover, after undergoing filtration and recrystallization processes, the purity of the BHET achieved was above 95.5% (Supplementary Figs. 34a and 35). Moving forward, the ΔBHETase hydrolysis step resulted in the production of TPA with up to 84.1% yield

(Supplementary Fig. 34b). These findings demonstrate the efficiency and effectiveness of the combined PET glycolysis and ΔBHETase hydrolysis processes for various plastic waste degradation and TPA production.

Motivated by the previous results and to complete a closed-loop PET recycling, we resynthesized a total of 554 mg virgin PET directly from the chemical-enzymatic degradation solution (920 mg of TPA) (Fig. 6a and Supplementary Fig. 36). And the entire cycle from degradation to polymerization can be completed within a few days (Fig. 6b and Supplementary Fig. 37a). These results confirmed the viability of producing virgin PET from non-petroleum sources using our tandem PET recycling system in a closed-loop manner. Besides, converting plastic waste into various high-value chemicals (e.g., fuels, waxes, and lubricants) offers an open-loop upcycling perspective to address the urgent issues of plastic waste. Herein, p-phthaloyl chloride, as a single exemplary chemical, was synthesized using the TPA yielding from plastic wastes to achieve an open-loop PET upcycling (Fig. 6c and Supplementary Fig. 37b, see more details in Supplementary Note 10). Collectively, leveraging the engineered ΔBHETase, we have successfully demonstrated a process concept that uses a BHETase-joined tandem PET degradation system to transform plastic waste into valuable products. These technologies hold potential in supporting a circular plastics economy, where plastic waste is efficiently recycled and repurposed.

## Discussion
This work reports the discovery and rational engineering of BHETases, guided by knowledge and mechanistic understanding. The two discovered BHETases from *Chryseobacterium sp.* PET-29 and *Bacillus subtilis* PET-86 offered a starting point for engineering enzymes that are capable of contributing to PET biodegradation. The robust engineered ΔBHETases (ΔChryBHETase and ΔBsEst) with easier substrate access showed increased catalytic efficiency and thermostability during BHET degradation. We envision that the approach of discovering and barrier-engineering BHETases demonstrated in this work could be applied to other PET hydrolases, like PETases, MHETases, and cutinases. Meanwhile, the ΔBHETase-joined tandem chemical-enzymatic approach was developed to achieve both closed-loop PET recycling and open-loop PET upcycling. The thermostable BHETases could be applied in the future to unlock in situ PET degradation by consuming the BHET directly after PET glycolysis. Beyond the virgin PET and p-phthaloyl chloride we have shown here, further chemical engineering could allow chemical process enhancements to access more significant titers, rates, and the manufacturing of other products. To guide future progress toward eco-friendly and cost-effective processes, techno-economic analysis and life cycle evaluation would be required.

## Methods
### Experimental section
**Materials.** All analytical grade or higher purity chemicals were purchased from Merck (Darmstadt, Germany) and Sigma-Aldrich Co. (St.

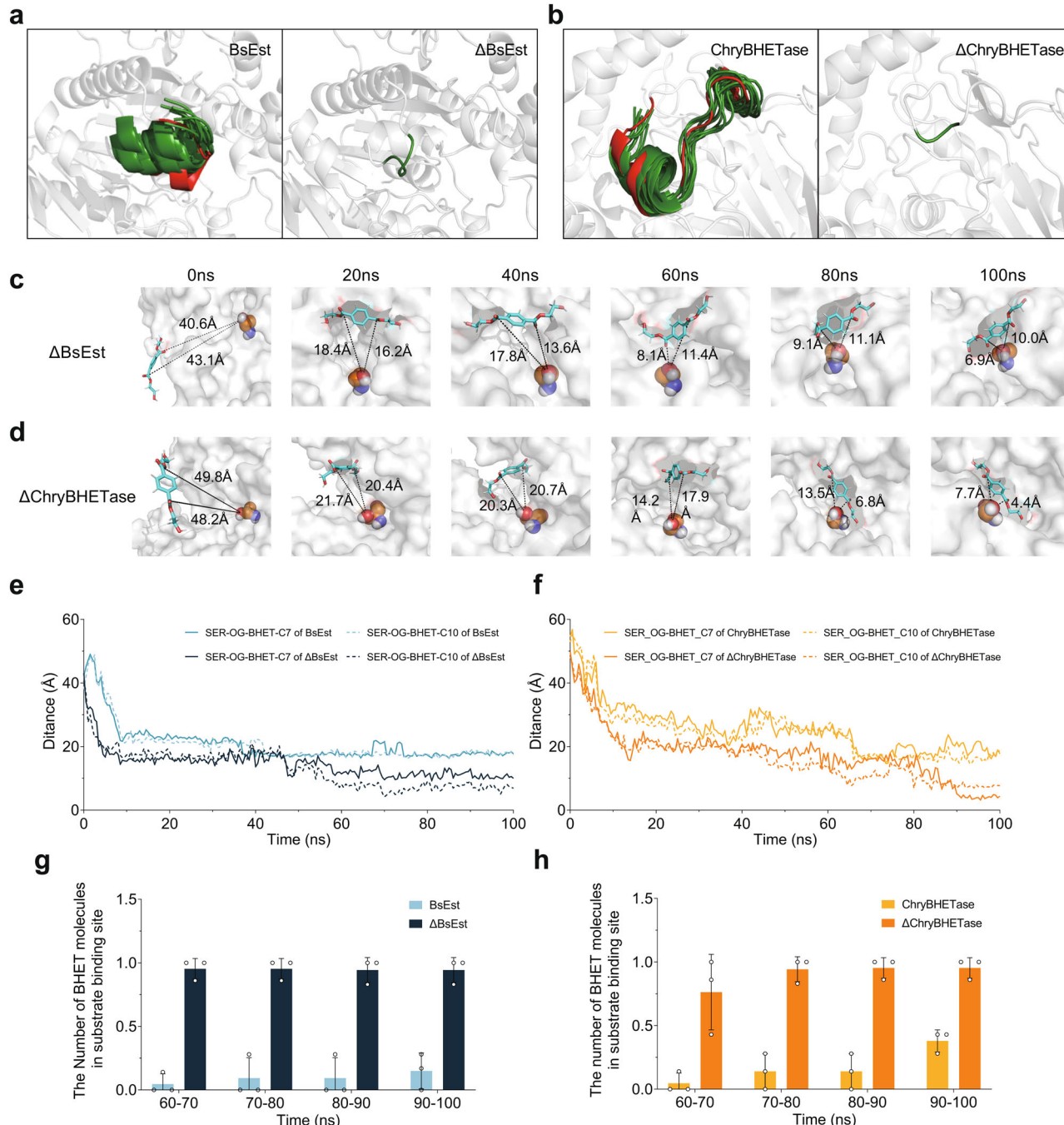

**Fig. 4 | Experimental verification of the improved truncated variants upon BHET and in silico analysis.** The spatial positions of the truncated region in **a** BsEst (left) and ΔBsEst (right) and **b** ChryBHETase (left) and ΔChryBHETase (right) were output every 10 ns during the simulation, with the region in the first frame indicated in red and the others in green. Changes in the position of the local structure of **c** ΔBsEst and **d** ΔChryBHETase between the active sites and BHET from 0–100 ns. The active site was shown to spheres, and the BHET molecule was shown to sticks. The distance curves of **e** BsEst and ΔBsEst and **f** ChryBHETase and ΔChryBHETase between the hydroxyl oxygen atom (OG) of Ser and the carbonyl carbon atoms (C7 and C10) of the BHET molecule at 100 ns simulation proceeding Data plotted from the average of three independent MD runs ($n = 3$). The number of BHET molecules in the substrate binding site of **g** BsEst and ΔBsEst and **h** ChryBHETase and ΔChryBHETase every 10 ns as a gradient in the last 40 ns of the simulation. Error bars showed the standard deviation (s.d.) from three independent MD runs for each simulation.

Louis, MO, USA) and used as received unless stated otherwise. The amorphous PET film was purchased from Goodfellow (UK, product code: ES303010) and cut into ⌀6 mm pieces (⌀ = 6 mm, roughly 67 mg) for further experiments.

**Screening microorganisms using PET powder as a sole carbon.** Fifty samples were collected from the landfill in Nanjing, China, including

PET bottles, PET boxes, PET film, PET sheets, and other debris and surrounding soil samples. For each flask, 2 g samples were added to 100 mL of solid medium containing diethyl phthalate (DET, PET structural analogs) as the sole carbon source. After shaking 150 r min⁻¹ at 30 °C for 5 days, 1 mL of each enrichment medium was added to a new 100 mL DET enrichment medium and incubated for 10 cycles under the same conditions. The 10th enrichment was diluted

**Table 1 | Kinetic parameters of BHETase and ΔBHETase[a]**

| Enzyme | $K_M$ (mM) | $k_{cat}$ (s$^{-1}$) | $k_{cat}/K_M$ (s$^{-1}$mM$^{-1}$) |
|---|---|---|---|
| BsEst | 0.1 ± 0.02 | 37.7 ± 1.98 | 382.11 ± 84.51 |
| ΔBsEst | 0.08 ± 0.03 | 102.2 ± 5.18 | 1350.77 ± 334.03 |
| ChryBHETase | 0.09 ± 0.02 | 6.85 ± 0.53 | 77.41 ± 11.04 |
| ΔChryBHETase | 0.12 ± 0.03 | 12.88 ± 1.24 | 109.19 ± 12.87 |

[a]All reactions were carried out in 50 mM potassium phosphate buffer (pH 7.5) at 30 °C using BHET as substrate. The values were averaged from three replicates ($n$ = 3) with the standard deviation (s.d.).

appropriately and applied to inorganic salt solid media containing PET powder at 30 °C for 5 days. Then, a single strain was picked onto the inorganic salt solid medium with PET powder and this step was repeated several times. Finally, the microorganisms adhered to PET powder were screened and purified (Supplementary Fig. 1).

The genome was extracted using the Bacterial Genome Extraction Kit (TaKaRa). The 16S rRNA gene was amplified by PCR using bacterial universal primers 27F (5′-AGAGTTTGATCCTGGCTCAG-3′) and 1492R (5′-TACGGCTACCTTGTTACGACTT-3′) (Tsingke Biotechnology Co., Ltd. (Nanjing, China)). The 16S rRNA gene sequences obtained were analyzed for homology in the NCBI database Blast, and strains with high homology were selected to construct a phylogenetic tree (Supplementary Fig. 2). The strain PET-29 was deposited in the China Center for Type Culture (CCTCC NO. 2022311). The GenBank accession numbers for the 16S rDNA gene sequence of strains are shown in the Supplementary 16S rDNA section.

**Whole-genome sequencing analysis of strain PET-29.** To obtain relevant gene annotations, the whole genome obtained above was split into two and sequenced using NovaSeq 6000 (Illumina, USA) and PromethION (Oxford Nanopore Technologies, Oxford, UK) respectively. The complete project process, including sample quality testing, library construction, library quality testing, and library sequencing, was carried out according to the standard protocols provided by the sequencing company (Tsingke Biotechnology Co., Ltd. (Nanjing, China)). Bioinformatics analysis includes the following major steps: filtering and counting of raw sequencing data, genome assembly, genome composition analysis, functional component analysis, functional annotation, and genome visualization. In brief, after filtering the splice, fragment and low-quality data, the data from the three generations of Nanopore sequencers yielded a total of 1,000,002,109 bp of clean data for assembly and 991,168,200 bp of clean data from the second generation illumina sequencing; the size of the assembled complete genome was 4.125 M. The assembled genome was predicted by Prokka (1.12) software, and the KEGG (VERSION: 87.0-r20180701), COG (VERSION: 2014), and GO (VERSION: 2014) databases were used for functional annotation.

**Potential enzymes predicted by Sequence Similarity Networks (SSN).** Sequence similarity networks (SSNs) are an effective way to visualize and analyze the similarity relationships between members of a superfamily[81]. SSNs facilitate the inspection of very large sets of protein sequences and enable the simultaneous assessment of orthogonal information. This study applied the EFI-EST tool for generating SSNs for protein families (https://efi.igb.illinois.edu/efi-est/)[82] with default parameters. Protein multiple alignment were conducted using Clustal Omega 1.2.2.

**Seamless cloning and construction.** *E. coli* DH5α was used as a host strain for plasmid construction. Target gene cloning was performed by designing primer probes using the NCBI database genome sequence. The primers used in this study are listed in Supplementary Table 12. All the enzymes used in this study were constructed into pET-22b(+) by

the Seamless Cloning Kit (Beyotime, China). Ligation between the target fragment and pET-22b(+) vector was incubated for 30 min at 50 °C, then adding 10 μL reaction mix to 100 μL of *E. coli* DH5α competent cells and rested for 30 min. They were coated on LB plates after heat shock for 60 s. All microorganisms and plasmids used in this study were complemented in the supplementary information.

**Construction of truncated enzymes.** The truncated enzymes were obtained by deleting the residues in the barrier region. ΔBsEst was constructed by deleting nine residues (L410-K418) from BsEst. ΔChryBHETase was constructed by deleting 11 residues (V66-K76) from ChryBHETase. The insert codons (-GG-) connecting the truncated junctions were underlined. After confirming the PCR product by DNA electrophoresis, the reaction mixture was digested with DpnI (37 °C, 3 h) and transformed (1 μL) into *Escherichia coli* DH5α (heat shock at 42 °C for 45 s), followed by expressing in *Escherichia coli* BL21(DE3).

**Enzyme expression and purification.** The recombinant plasmids above were transformed into *Escherichia coli* BL21 (DE3). The recombinant bacteria were inoculated into Luria Bertani (LB) medium overnight at 37 °C/200 r.p.m. with proper antibiotics. The 1 mL of seed cultures were incubated in 100 mL LB media at 37 °C/200 r.p.m. When the optical density of cultures at 600 nm (OD$_{600}$) reached 0.4–0.6, isopropyl-1-thio-β-D-galactopyranoside (IPTG) was added to final concentrations of 0.1 mM. After induction, the bacteria were cultivated at 20 °C/150 r.p.m. for 24 h.

*E. coli* cell pellets were harvested by centrifugation at 8000 × g for 10 min at 4 °C and resuspended in 50 mM Na$_2$HPO$_4$-NaH$_2$PO$_4$ buffer (pH 7.5) after washing twice with buffer (pH 7.5). The washed cells with 1 mg/mL lysozyme were disrupted by sonication (3-sec pulse on, 5-sec pulse off, AMP 35%). Cell debris was removed by centrifugation at 4 °C, 8000 × g for 1 h, and the supernatants were filtrated by a 0.22 μm filter (Choice filter, Thermo Scientific).

To obtain the purified enzymes, the supernatant was incubated with a Ni-affinity resin (Sangon Biotech, China) for 1 h. It was washed 5 times with the same volume of washing buffer (50 mM sodium phosphate buffer, 300 mM sodium chloride, and 20 mM imidazole, pH 8.0). Then, the affinity resin was eluted with elution buffer (50 mM sodium phosphate buffer, 300 mM sodium chloride, and 250 mM imidazole, pH 8.0). The purified enzymes were concentrated using a 10 kDa Amicon Ultra centrifuge tube (Millipore, Burlington, USA). Before the enzyme reaction, the enzyme concentration was quantified by the BCA Protein Assay Kit. The molecular weights were determined using sodium dodecyl sulfate-polyacrylamide gel electrophoresis (SDS-PAGE) (Supplementary Fig. 6).

**Enzyme activity and kinetics Assays for BHET.** The enzyme activities were determined using BHET as a substrate by a previously reported method with slight modifications[83]. Briefly, 20 μL of the purified enzyme was added into 1980 μL reaction buffer (50 mM Na$_2$HPO$_4$-NaH$_2$PO$_4$, pH 7.5) containing 5 mM BHET substrate to initiate the catalytic reaction. One enzyme unit (1 U) was defined as the amount of enzyme that converts 1 μmol BHET per minute at 30 °C. The supernatant was obtained by centrifugation (8000 × g, 10 min). After 0.22 μm filtration, 20 μL of assay solution was analyzed using a high-performance liquid chromatography system (HPLC, Agilent 1200 and Ultimate 3000 UHPLC systems) equipped with a Welch Ultimate XB-C18 column (4.6 × 250 mm, 5 μm, Welch Materials, Inc., Shanghai, China). The mobile phase was 40% methanol with 0.12% acetic acid (pH 2.5) at a flow rate of 0.7 mL min$^{-1}$, and the effluent was monitored at a wavelength of 240 nm.

The enzymatic reaction kinetics were determined by measuring the initial enzyme activity at substrate (BHET) concentration ranging from 0.8 to 15 mM at 30 °C. The initial rate was calculated using linear hydrolysis and plotted against BHET concentrations. All the

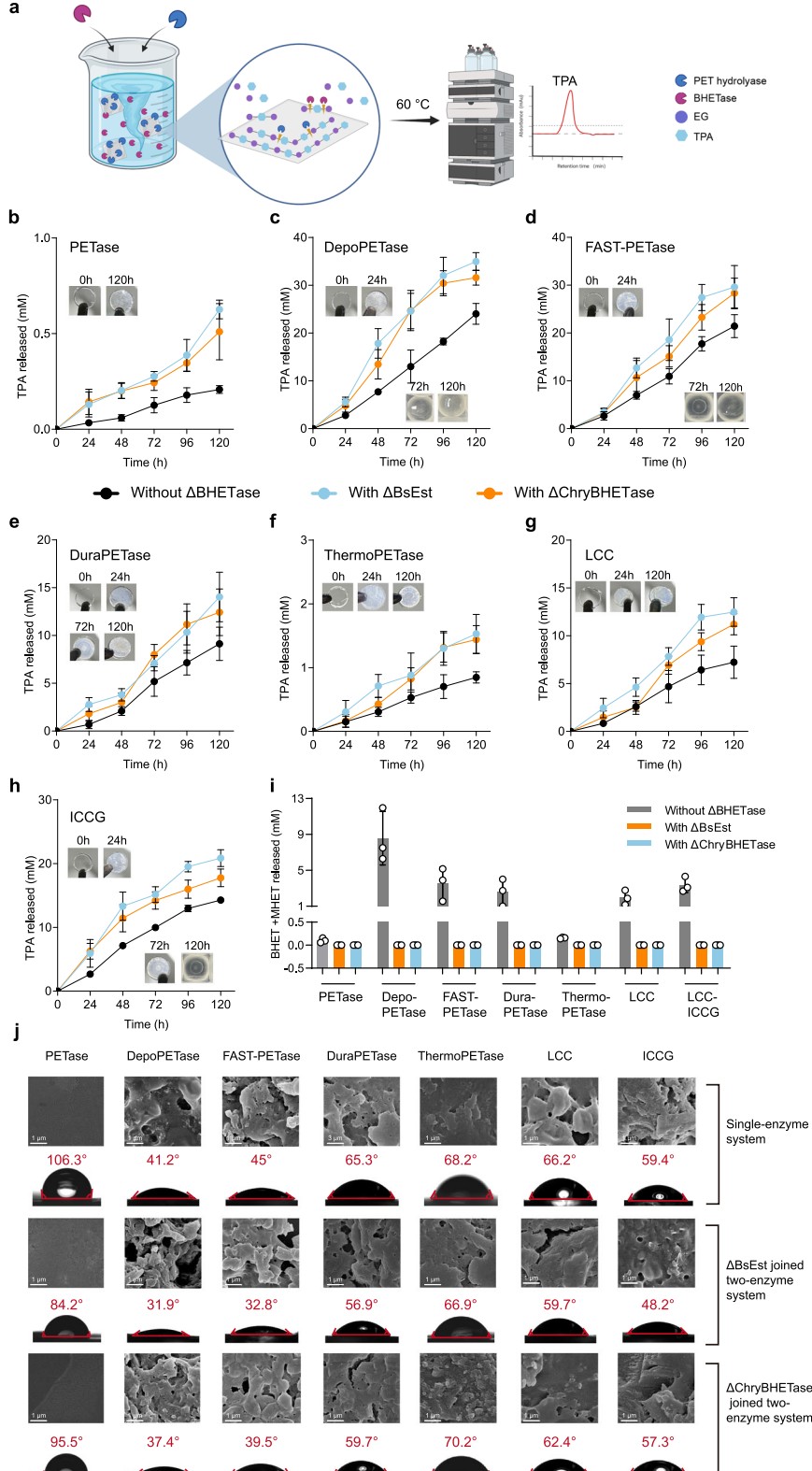

**Fig. 5 | Two-enzyme degradation system at 60 °C. a** An overview of the two-enzyme degradation system. Icon graphics of this figure was created by BioRender.com. Time course of PET depolymerization in a two-enzyme system with different PET hydrolases, including **b** PETase, **c** DepoPETase, **d** FAST-PETase, **e** DuraPETase, **f** ThermoPETase, **g** LCC, and **h** LCC-ICCG. Error bars correspond to the standard deviation (s.d.) of three measurements ($n = 3$). Reaction condition: The PET films (ø=6 mm, roughly 67 mg) were soaked in 2940 μL of Na$_2$HPO$_4$-NaH$_2$PO$_4$ (pH 8.5, 50 mM) buffer at 60 °C with 50 μL of 0.5 mg/mL PET hydrolases and ΔBHETases, respectively. The fresh enzyme solution was supplemented every 24 h for depolymerization. **i** The PET monomers (the sum of BHET and MHET) were released at 96 h. Error bars correspond to the standard deviation (s.d.) of three measurements ($n = 3$). **j** The SEM images (up panel) and water contact angle analysis (down panel) of the PET film in a single-enzyme degradation system, two-enzyme degradation system with ΔBsEst, and a two-enzyme degradation system with ΔChryBHETase after 48 h at 60 °C. Results were reproduced three times independently; representative photography and micrographs are shown.

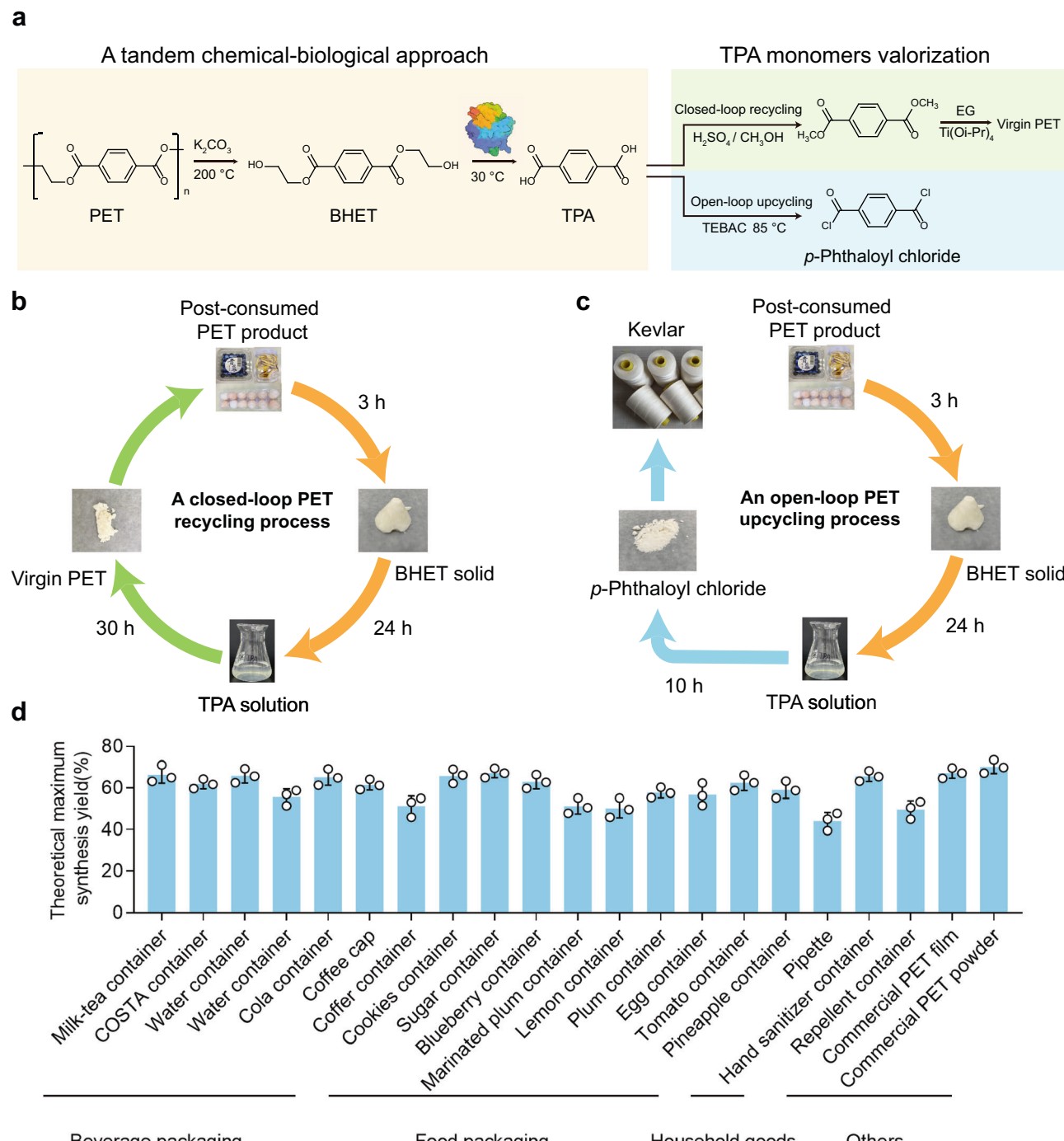

**Fig. 6 | Closed-loop recycling and open-loop PET upcycling based on the BHETase-joined tandem chemical and biological PET degradation system.** **a** The schematic of the closed-loop PET recycling and open-loop PET upcycling pathways. TEBAC: benzyl triethylammonium chloride. Icon graphics of this figure was created by BioRender.com. **b** The schematic of the closed-loop PET recycling. **c** The schematic of the open-loop PET upcycling. **d** Theoretical maximum synthesis yield of *p*-phthaloyl chloride from 21 commercial pc-products. Chemical glycolysis conditions: glycolysis of 5 g PET, 3 h of reaction time, 200 °C of temperature, and 0.1 g of potassium carbonate loading. Enzymatic catalysis condition: 2 mL of 0.5 mg/mL ΔBsEst and ΔChryBHETase hydrolyzing the BHET obtained from PET glycolysis at 30 °C within 24 h, respectively. Error bars correspond to the standard deviation (s.d.) of three measurements (n=3).

experiments were conducted in triplicate, and the mean values with standard deviations (s.d.) were reported.

**Determination of thermostability and half-life.** To determine the thermostability, the purified enzyme was incubated from 30 °C to 60 °C at intervals of 10 °C for 2 h, and then its remaining enzyme activity was determined as before. The relative enzyme activity was calculated using the enzyme activity at 30 °C as a control.

The purified proteins were stored separately at 50–70 °C (at 5 °C intervals), and the remaining enzyme activity was tested every 30 min until it was reduced to half of the initial enzyme activity. Based on the enzyme inactivation equation:

$$[E_t] = [E_0]\exp(-kt) \tag{1}$$

$E_t$: relative enzyme activities at time t; $E_0$: initial relative enzyme activities. Thus, half-life

$$t_{1/2} = \ln2/k \tag{2}$$

**The degradation of PET film and BHET.** Commercially sourced PET films (Goodfellow, product code: ES303010) were washed with 1% SDS for 30 min, followed by deionized water and ethanol, and then air dried overnight for enzymatic digestion experiments. Purified enzymes were incubated with PET film (⌀6 mm) in a buffer containing 50 mM $Na_2HPO_4$-$NaH_2PO_4$ (pH 7.5) at 30 °C for 24 h or more. The reaction was terminated by adding methanol. The supernatant was then collected and analyzed by HPLC. When using BHET as the reaction substrate, purified enzymes were incubated with 5 mM BHET in a buffer containing 50 mM $Na_2HPO_4$-$NaH_2PO_4$ (pH 7.5) at 30 °C. Samples were taken regularly at different time points. The SEM images and water contact angle were determined by Scanning Electron Microscope (SEM S4800) and Contact Angle Meter OCA20, respectively.

**Differential scanning calorimetry (DSC).** DSC (NETZSCH DSC 214) was used to determine the percent crystallinity of the PET films collected from post-consumer (pc) PET products. The following protocol was used for each sample. A sample of 2, 3 mg in an aluminum pan was cooled from room temperature to 0 °C. The pan was then heated from 0 to 300 °C at 10 °C min⁻¹ and then maintained at 300 °C for 1 min under a nitrogen atmosphere. Subsequently, the sample was quenched to 0 °C at 10 °C min⁻¹. The heat of fusion $\Delta H_m$ and cold crystallization $\Delta H_c$ were determined by integrating areas (J g⁻¹) under peaks. The percent crystallinity was calculated using the following equation:

$$crystallinity(\%) = \left[\frac{\Delta H_m - \Delta H_c}{\Delta H_m}\right] \times 100 \tag{3}$$

where $\Delta H_m$ is the enthalpy of melting (J g⁻¹), $\Delta H_c$ is the enthalpy of cold crystallization (J g⁻¹), and $\Delta H_m$ is the enthalpy of melting for a 100% crystalline PET sample, which is 140.1 J g⁻¹[,35,39].

**Chemical pretreatment of commercial post-consumed PET products.** All reagents for the experiment were obtained from commercial suppliers, and details are described in Supplementary Table 11. According to Kim et al. [75]., the glycolysis reactions of PET were set in a 250 mL round flask equipped with a flux condenser and magnetic stirrer. 5 g of PET pieces, 20 g of EG, and different amounts of potassium carbonate were added to the reaction flask. The reaction mixtures were heated to 210 °C for 3 h. After the glycolysis reaction, 400 mL of hot water was added to the mixture, and PET oligomers were removed by filtration. The filtrate was concentrated to 30–40 mL by rotary evaporation at 60 °C. The concentrated mixture was recrystallized in a refrigerator at 4 °C for 18 h. The BHET purity and BHET yield rate from 21 commercial pc products are calculated according to Eqs. (4) and (5), respectively. The theoretical weight of BHET is calculated by dividing the initial weight of 21 samples by the repeating unit (TPA-EG, 192.2 g mol⁻¹) and multiplying the molecular weight of BHET (254.24 g mol⁻¹). The theoretical maximum hydrolysis yield is calculated by dividing the actual molar concentration of TPA formed by the chemo-enzymatic depolymerization reaction by the theoretical maximum concentration of TPA produced from 21 pc-products (Eq. 6).

$$BHET\ purity(\%) = \frac{BHET\ peak\ area\ via\ HPLC}{Total\ peak\ area\ via\ HPLC} \times 100 \tag{4}$$

$$BHET\ yield(\%) = \frac{Actual\ weight\ of\ BHET\ after\ PET\ glycolysis[g]}{\frac{The\ initial\ weight\ of\ 22\ samples}{The\ repeating\ unit(TPA-EG,192.2\,g\,mol^{-1})} \times BHET\left(254.24\,g\,mol^{-1}\right)} \times 100 \tag{5}$$

$$Theoretical\ maximum\ hydrolysis\ yield(\%) = \frac{Actual\ molar\ concentration\ of\ TPA\ (m)}{\frac{Concentration\ of\ PET(g/l)}{The\ repeating\ unit(TPA-EG,192.2\,g\,mol^{-1})}} \times 100 \tag{6}$$

**Regeneration of virgin PET using TPA as a substrate.** BHET obtained from chemical glycolysis was serially treated by ΔBsEst and ΔChryB-HETase in 50 mM $Na_2HPO_4$-$NaH_2PO_4$ buffer (pH 7.5) at 30 °C. 2 mL of fresh enzyme (0.5 mg/mL) was added to the degradation solution for 24 h to maximize the enzymatic degradation rate. After degradation, the enzyme degradation solution was filtered, and the filtrate was collected for regeneration. The pH of the solution was subsequently adjusted to 2 with 37% HCl. The precipitate was filtered, washed several times with deionized water, and dried under vacuum overnight. Then TPA was collected and used in the next step without further purification.

TPA obtained from chemical-enzymatic catalysis was dissolved in $CH_3OH$ at room temperature, and 95% $H_2SO_4$ was added dropwise to the suspension, after which the mixture was stirred under reflux for 24 h to become a clear solution and cooled to room temperature. The reaction mixture was recrystallized from dimethyl terephthalate (DMT), washed with $CH_3OH$, and dried under vacuum to obtain white powder. The product (DMT) was then assayed by LC-MS.

DMT (1.39 g) was then added to the vacuum reactor with EG (1.20 mL) and titanium isopropoxide (0.04 mL). The reaction was stirred at 160 °C, 200 °C, 210 °C for 1 h, 1 h, 2 h, respectively, and the final temperature was further increased to 260 °C for 2 h, and then cooled to room temperature. The resulting PET was dissolved in 1,1,1,3,3,3-hexafluoro-2-propanol (HFIP, 10 mL) and added dropwise to $CH_3OH$ to remove the catalyst. The PET (0.63 g) was collected as a white solid after centrifugation and dried under vacuum. The crystallinity of virgin PET was detected by DSC.

**Synthesis of *p*-Phthaloyl chloride using TPA as a substrate.** A total of 1.5 g of TPA, 4.4 g of thionyl chloride, and 0.128 g of benzyl triethylammonium chloride (TEBAC) were taken in a 500 mL three-necked flask. One-third of thionyl chloride was added to the flask, then stirred magnetically for 20 min at room temperature. After slowly adding the remaining thionyl chloride, the reaction mixture was heated to reflux at 85 °C for 4 h. The reaction was distilled under reduced pressure to recover thionyl chloride, cooled, and crystallized. 10 mL of petroleum ether was added for recrystallization when a slight solid precipitated. *p*-Phthaloyl chloride was collected as white solids after centrifuging and dried under vacuum.

**Computational section**

**In silico modeling and analysis.** The AlphaFold was used for predicting the BsEst and ChryBHETase with each of the five trained model parameters[84]. The MSA generation, AlphaFold predictions, and structure relaxation with Amber were run on the local server with GPU. A full database (updated to 2021-09-17) was applied for all structure predictions.

For the docking of the substrate (BHET, MHET, TPA, and 2PET) to all enzymes obtained in this study and their corresponding truncated enzymes, the structures generated from the above AlphaFold were used as the receptors for the model organization. The receptors and ligands (BHET, MHET, TPA, and 2PET) were constructed and parameterized using AutoDock Tools 1.5.6. The predicted catalytic residues were used to define the binding pocket, and clustering analysis of the output results was conducted by AutoDock using criteria such as energy minimization and cluster size. The energetically favorable poses of ligands binding to the targeted sites of enzymes were extracted and analyzed. The docking results with several most favorable conformations were then visualized and analyzed by Pymol 2.5.2.

 

The normalized scoring was obtained by assigning weights of 10%, 30%, 40%, and 20% to the binding energy, the number of hydrogen bonds, distance, and volume, respectively.

$$\text{The normalize scoring} = \left(1 - \frac{A1 - A\min}{A\max - A\min}\right)*10\% + \frac{B1 - B\min}{B\max - B\min}*30\% + \left(1 - \frac{C1 - C\min}{C\max - C\min}\right)*40\% + \frac{D1 - D\min}{D\max - D\min}*20\%$$

(7)

Among them, A represents the binding energy of different receptor proteins to the ligand (BHET) calculated by molecular docking; B represents the hydrogen bonds generated with surrounding amino acid residues; C represents the distance between the oxygen of the Ser hydroxyl group and the carbonyl carbon (C7 and C10) of the BHET molecule, respectively; and D denotes the cavity volume formed between BHET and the amino acid residues within 4 Å.

**Molecular dynamics simulation.** Molecular Dynamics (MD) simulation and analysis were performed with the GROMACS 2016 simulation package with the GROMOS96 (54a7) force field. The validated BHET model was taken from the ATB website. A total of 24 simulations were performed with BsEst and ChryBHETase in the pure water system and in water systems with the addition of BHET molecules under the experimental condition. Similarly, the truncated variants ΔBsEst and ΔChryBHETase were performed in both systems. ΔBsEst and ΔChryBHETase were the optimal conformations obtained by AlphaFold structure prediction and optimization. The structures BsEst, ΔBsEst, ChryBHETase, and ΔChryBHETase were solvated into a cubic box with a minimal distance of 10 Å from the box edge to the protein. Depending on the experimental concentration, it was determined that 30 BHET molecules were added into the box, and the system was filled with the SPCE water model. Na$^+$ and Cl$^-$, as counterions, were used to neutralize the total net charge of the systems to achieve a net charge of zero. To avoid the most unfavorable interactions, energy minimization using the steepest descent method was performed before MD simulations. The 100 ps NVT as the first and then 100 ps NPT ensembles were performed at temperatures close to 298 K, followed by a production simulation run (100 ns at 298 K, 1 bar, and a time step of 2 fs). Three independent MD runs were performed with different starting atomic velocities to evade the dependence of the simulation results on the beginning parameters. The coordinates, energy, and velocity were stored every 0.5 ns for trajectory analysis. We used GROMACS simulation package tools to calculate all the analyses, such as the root-mean-square deviation (RMSD), solvent-accessible surface area (SASA), radius of gyration (R$_g$), root-mean-square fluctuation for each residue (RMSF), spatial distribution function (SDF), solvation phenomenon, et al. Simulated trajectories were visualized and analyzed using Pymol 2.5.2 and VMD 1.9.3.

**Statistics and reproducibility**

GraphPad Prism 8 version 8.4.3 (GraphPad Software) were used to analyze the data. All bar graphs represent mean and standard deviation (s.d.) and each data point represents one independent biological replicate. Number of biological replicates are indicated in the figure legends, at least 3 replicates were used for all experiments. The source data underlying Figs. 1–6, Supplementary Figs. 2–5, 7–9, 11–20, 24, 25, 29, and 32–34 are provided as a Source Data file. Source data are provided with this paper.

**Reporting summary**

Further information on research design is available in the Nature Portfolio Reporting Summary linked to this article.

## Data availability

The strain *Chryseobacterium sp*. PET-29 is preserved in China Center for Type Culture Collection (CCTCC), and the accession code is CCTCC NO: M 2022311. The 16S rRNA of reported strains have been deposited with GenBank under accession codes as follows: *Bacillus subtilis* PET-86 under accession code OP564167"; *Chryseobacterium sp*. PET-29 under accession code OP564169. The assembled genome for *Chryseobacterium sp*. PET-29 has been deposited under accession code CP107053, with the raw sequencing data available under accession code PRJNA886060. The nucleotides of the reported enzyme have been deposited in the GenBank under accession codes as follows: ChryBHETase, WP_263602248.1. The source data underlying Figs. 1–6, Supplementary Figs. 2–5, 7–9, 11–20, 24, 25, 29, and 32–34 are provided as a Source Data file. Source data are provided with this paper.

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

## Acknowledgements

This work was supported by the National Key Research and Development Program of China under grant no. 2019YFA0706900 (H.H.), the National Natural Science Foundation of China under grant no. 22008123 (X.L.), and Jiangsu Key Laboratory for Numerical Simulation of Large-Scale Complex Systems. Icon graphics of this work were supported by BioRender.com.

## Author contributions

H.C. and X.L. conceived of the presented idea. A.L. carried out the experiments. Y.J. S. performed molecular dynamics simulations. A.L., Y.J.S., and H.C. analyzed and visualized the data. H.C., A.L., and Y.J.S. wrote the manuscript with input from all authors. M.W., L.W., Y.B.S., and R.Y. carried out part of PET degradation experiments. X.L. and H.H. provided supervision and resources for this study.

## Competing interests

The authors declare no competing interests.
