## [Peer Review File · Nature Communications]

Discovery and mechanism-guided engineering of BHET hydrolases for improved PET recycling and upcyclingReviewers' comments:

Reviewer #1 (Remarks to the Author):

The authors present a study on the identification of BHETase enzymes and demonstrate a utility in utilizing these compounds to convert into TPA. At the onset, this topic is of importance and sounds necessary in the field. However, the paper is poorly written and the overall impact is highly diminished by the experimental conditions and analysis.

Most importantly, there are many PETase enzymes that are able to achieve much higher release of TPA and even enable full degradation of PET. These enzymes typically operate in a 50-70C range (unlike the 30C range used in this study). Properly coupling the PETase with a BHETase explored here would demonstrate if there is any true utility to the work here. Likewise, there is a lack of full degradation in this study. This is another important factor as there are more challenges for PETase and MHETase at these higher degrees of degradation (as crystallinity increases and BHET/MHET increase over the timecourse of an experiment). On the flip side, many higher temperature PETase enzymes in the literature have been able to demonstrate full conversion of PET to TPA (not getting stuck at BHET or MHET as the authors suggest). It is not clear the authors are solving an "critical" problem here compared to how they are portraying it.

It is somewhat odd that the authors have chosen to bypass the PETase all together and adopt a chemical approach that then produces BHET. The authors have not fully justified this approach, have not fully compared efficiency head-to-head here with a two enzyme system. In fact, much of the text almost alludes to / builds up to a two enzyme system and this is not completed in the end.

While the authors showcase production of p-phthaloyl chloride as an exemplar product (likely due to its appearance in products like Kevlar), this will (1) not solve the plastics issue as the demand for this molecule is far less than the amount of PET produced and (2) this does not create a circular economy with respect to plastics / PET.

Error bars are never really described here as to their meaning and replicates.

Reviewer #2 (Remarks to the Author):

General comments:

The paper provides some new information on BHET hydrolyzing enzymes. Despite these novel insights, the paper lacks a solid comparison with the many existing PET hydrolyzing enzyme systems (identical reactions conditions, substrates etc).

Specific comments:

Why was diethylterphthalate used?

l117 onwards, very difficult to read, maybe put all references to figures at the end of sentences

l 124 please rewrite, all PET/BHET hydrolyzing enzymes are esterase, is there a new EC enzyme number for BHETases?

Legend Fig 2e „ upon BHET -> upon incubation on BHET

Other longer-chain oligomers should be assessed as substrates for the enzyme

l199 Discuss kinetic parameters, 100% conversion after a certain does not tell much

l 212 provide half life times at different temperatures rather than percentual values for arbitrary time points

l294 what does highest activity towards amorphous PET mean? There are enzymes that are much more efficient, provide clear numbers for such statements

l 303 the activity of the enzyme system should be compared in detail with the activity of other published polyester degrading enzymes such as the LCC at identical conditions.

I 316 Efficient manner, - see above, a detailed comparison with the many other enzyme and enzyme systems published for PET hydrolysis is required

Fig 6b Plastic products hydrolysis, this would only be valuable when compared to other existing PET hydrolyzing enzymes and when detailed composition / properties (e.g. % crystallinity) of products would be given

Reviewer #3 (Remarks to the Author):

Comments to manuscript NCOMMS-23-02054

Li et al. describe in this manuscript the discovery of two enzymes (named ChryBHETase and BsEst) which can hydrolyze the compound BHET, a water-soluble intermediate formed in the degradation of PET (using PETases able to act on the large polymer), which is very similar to MHET for which so-called MHETase have already been described in literature (the only difference between MHET and BHET is the presence of an additional ethylene glycol unit present at the terephthalic acid).

Technically, the authors have used an entire classical approach (shown in Figure 1, b) to identify these two hydrolases: take environmental microbial samples, tests them for the desired activity, identify the microorganism and then the gene encoding the "BHETase". Produce them recombinantly, purify and biochemically characterize them followed by improving them using typical methods for protein engineering. In addition, they have predicted the structures of both enzymes using AlphaFold2 and compared them to known enzymes active on BHET and/or MHET. Even if they name their approach "mechanism-guided barrier engineering" and name a method "KnowCover" strategy, this does not mean that the concepts are novel at all. Researchers perform mechanism-guided enzyme engineering using computational tools and directed evolution since about two decades. As stated above, the authors have used very common approaches to identify novel enzymes for applications in biocatalysis (not only for plastic degradation) and improved them using state-of-the-art methods.

It was not a true pleasure to read the manuscript as the English is not great and extensive language editing is required. Furthermore, many scientific aspects are vague, not clear or not put into the right context (with respect to literature data/reports) as outlined below. Nevertheless, it is an interesting study which in principle deserves publication after thorough revision, however, the concepts/methods used and achievements made are not outstanding.

Major comments:

In the introduction they claim that although many microbes and enzymes have been described for PET degradation (which is correct), ‘researchers are still mining for the last piece of the puzzle’, the BHETase. This was for sure their motivation, but in fact BHET does not represent a major problem in the enzymatic recycling of PET. As shown in the two publications by Tournier et al. (Nature 2020, Ref. 35) and the Alper team (Nature 2022, Ref. 37), which represent the ‘bench-mark’ processes, BHET is hardly detectable during hydrolysis of PET. Even MHET (which has been reported in a few publications to maybe inhibit the PET-hydrolyzing enzyme) does not present a major problem in large scale processing of PET (so this ‘last piece of puzzle’ is not as relevant as they believe/write in the manuscript). It is stated on page 4, line 71, that in the process developed by Tournier et al. using LCC (note: in that process the highly active and thermostable variant LCC-ICCG, is used not LCC wildtype) “a minimum of 90% conversion” is achieved. In fact, they achieve >98% conversion at 200 g/L, which in turn indicates that BHET (or MHET) are hardly present at all during the process and get rapidly hydrolyzed. The same holds true for the many examples reported by the Alper team in their paper (Ref. 37), where also complete conversion of PET was published (without adding a BHET or MHETase).

Abstract and main text: they write that the BHETases show up to 3.7-fold enhanced catalytic efficiency and thermostability: Firstly, what has been improved 3.7-fold? Activity or thermostability or both? That makes no sense as both features can’t be put into one number of fold improvement. Secondly, 2.7-fold compared to what? The wild-type enzymes? In addition, this number does not tell us much if we don’t know how many grams or moles / liter of BHET are hydrolyzed per hour (or minute). It is important that if a BHETase is used in a PET-hydrolysis process in combination with an efficient PETase (as those mentioned in the Nature papers and in their own study), that the BHETase is as efficient in its turnover as the PETase, otherwise such an enzyme is not useful.

The same “number problem” is true for the “7.0-fold increase” stated in line 30 of the abstract. 7.0-fold increase compared to the process from Tournier? They convert 200 g/L within 10 h, so 7.0-fold increase would be 1.400 g/L in 10 h or 200 g/L within 1.6 hours? I seriously doubt that productivity. To claim in line 31 that their BHETase boosts “state-of-the-art” PETase is not supported by any of their experimental data in comparison to the bench-mark processes (see above). In Figure 3d, substrate concentrations are given as 5 mM BHET (and mM BHET/PET in Figure 5 when the two enzymes were used together), this is orders of magnitudes (!) below the concentrations of PET used in the process by Tournier et al. and the Alper team, which both set the bench mark for efficient PET recycling. When they use minimum 200 g/L, the corresponding concentration of BHET formed is significant (if the authors of this paper are right that BHET represents a problem in the process, which I doubt, see above).

For a better comparison with the benchmark PETases, they should better report the activity of the BHETase as gram TPA formed per liter and hour and provide the specific productivity (e.g., TPA formed per liter and hour and g enzyme).

Consequently, phrases like “the excellent BHET degradation performance...” are wrong and misleading.

The authors could improve the thermostability of the BHETase, but the best temperature values studied/achieved appear still too low to enable a combination with the PETase designed by Tournier et

al. (Ref. 35, process runs at 70°C to overcome the crystallinity of PET) or by the Alper team (Ref. 37, process runs around 50°C). to achieve complete PET hydrolysis.

Synthesis of p-Phthaloyl chloride: this is not of much interest and not new. There are many possible chemical reactions to be done with terephthalic acid (TPA) obtainable from PET hydrolysis. To make this halogenated precursor does not fit at all into this manuscript. Moreover, I believe that it would be much better for the climate, if the TPA obtainable from PET hydrolysis is used to make new PET polymer (as demonstrated in Ref. 35 and 37) so that no new petrol must be used to make PET, rather than doing old-fashioned chlorine chemistry.

Other comments:

The enzymes were found in typical microbes known to produce carboxylesterases (especially *Bacillus* species are known to make esterases very useful for various biocatalytic applications). Thus, it is not unlikely to find an esterase in a *Bacillus* strain, which can also act on the rather simple water-soluble ester BHET. To call them now BHETase is in principle correct as they used BHET as substrate, found activity and added 'ase' to the compound. Most likely the "BHETase" can act on numerous other esters.

They often misuse the terms "degradation" and "recycling". Recycling is definitely preferred as the monomer building blocks can be accessed and used to make new polymer. This has been achieved for PET (e.g., by the Tournier et al. and Lu H. et al. teams, Ref. 35 and 37). Degradation does not imply that the monomers can be accessed to make new plastics, also this is impossible for e.g. polyethylene or PVC, where degradation can only yield low molecular weight compounds, but for sure neither ethylene nor vinylchloride as the double bonds reacted in the polymerization). Thus, the use of both terms "degradation" and "recycling" should be carefully checked in future manuscript versions. When using microbes (instead of isolated enzymes), even for PET recycling is unlikely as the products released can be metabolized by the microbes, which is rather a downside.

The original comments of the reviewers are in italics.

Manuscript Number: NCOMMS-23-02054

Manuscript Title: Mechanism-guided barrier engineering to boost the BHET hydrolases discovered from the environment into PET recycling and upcycling

Reviewer #1:

General Comments to the Authors: *The authors present a study on the identification of BHETase enzymes and demonstrate a utility in utilizing these compounds to convert into TPA. At the onset, this topic is of importance and sounds necessary in the field. However, the paper is poorly written and the overall impact is highly diminished by the experimental conditions and analysis.*

We greatly thank you for commenting that “*this topic is important and sounds necessary in the field*”. We carefully revised the entire manuscript, including proofreading for language and readability improvements. To prove the significance of our work, we included a comparison of seven state-of-the-art PET hydrolyzing enzyme systems that have been previously reported, including FAST-PETase¹, DuraPETase², ThermoPETase³, DepoPETase⁴, LCC and LCC-ICCG⁵ (Please refer to **Results 2.4** in the revised version).

Comment 1:

Most importantly, there are many PETase enzymes that are able to achieve a much higher release of TPA and even enable full degradation of PET. These enzymes typically operate in a 50-70 °C range (unlike the 30 °C range used in this study). Properly coupling the PETase with a BHETase explored here would demonstrate if there is any true utility to the work here. Likewise, there is a lack of full degradation in this study. This is another important factor as there are more challenges for PETase and MHETase at these higher degrees of degradation (as crystallinity increases and BHET/MHET increase over the time course of an experiment). On the flip side, many higher temperature PETase enzymes in the literature have been able to demonstrate full conversion of PET to TPA (not getting stuck at BHET or MHET as the authors suggest). It is not clear the authors are solving an “critical” problem here compared to how they are portraying it.

It is somewhat odd that the authors have chosen to bypass the PETase all together and adopt a chemical approach that then produces BHET. The authors have not fully justified this approach, have not fully compared efficiency head-to-head here with a two enzyme system. In fact, much of the text almost alludes to / builds up to a two enzyme system and this is not completed in the end.

Explained and addressed.

We fully agree that coupling the PETase with the explored BHETase can demonstrate the necessity of involving the BHETase in the PET degradation process. Besides the existing results (PETase-BHETase, and PETase- Δ BHETase at 30 °C), we investigated a two-enzyme degradation system that coupling more PET hydrolases coupling with Δ BHETase (Section 2.4, Fig. 5a). In total, seven state-of-the-art PET hydrolase were selected for comparison, including PETase¹, DepoPETase⁴, FAST-PETase⁶, DuraPETase², ThermoPETase³, LCC⁷ and LCC-ICCG⁵. To explore the feasibility of Δ BHETase at high temperatures, a representative reaction condition at 60 °C was selected. This temperature choice aligns with the commonly utilized conditions in several PET degradation studies^{4-6,8}. To comprehensively evaluate the PET degradation performance, we monitored multiple parameters, including total PET release, pure TPA production, BHET production, and MHET production. These measurements provide a comprehensive assessment of the efficiency and effectiveness of the PET degradation process, ensuring a complete understanding of the overall performance and the specific outcomes of the degradation system.

As shown in Fig. 5b-h, coupling Δ BHETase with all seven PET hydrolases (two-enzyme system) significantly shifted the TPA production to a much higher level than using PET hydrolase only (single-enzyme system). Interestingly, we did not observe complete hydrolysis of PET films to homogeneous TPA for all single PET hydrolases within 120 h under current depolymerization conditions (Fig. 5b-h). These results are different from previous studies on FAST-PETase⁶ and LCC-ICCG⁵. Unfavorable circumstances and/or the crystallinity of plastics could be the root causes^{9,10}. In contrast, the incorporation of Δ BHETases into a single depolymerization system at 60 °C resulted in the complete hydrolysis of PET films to TPA within 120 h, as observed in DepoPETase- Δ BHETase and FAST-PETase- Δ BHETase. Additionally, both Δ BHETases (Δ BsEst and Δ ChryBHETase) significantly enhance the overall efficiency of PET degradation within the initial 48 h. (Fig. 5b-h). Δ BsEst, for instance, helped DepoPETase and FAST-PETase improve by 1.8- and 2.0-folds, respectively, in comparison to employing PET hydrolase alone. These results demonstrate that incorporating BHETase into the PET degradation process leads to a substantial acceleration of PET degradation and a reduction in degradation time.

Furthermore, we conducted half-life studies on BHETase and Δ BHETase to demonstrate the thermostability of Δ BHETase in effectively hydrolyzing PET under high-temperature conditions. As indicated in Table S9, both BHETase and Δ BHETase maintained substantial activity across the temperature range of 50°C to 60 °C. Additionally, the engineered Δ BHETase, particularly Δ BsEst, exhibited improved thermostability compared to the wild-type variant.

In previous studies^{2,4-6,8}, researchers have often chosen to utilize high temperatures due to their ability to reduce the crystallinity of PET. This reduction in crystallinity facilitates plastic decomposition, making it more susceptible to degradation processes. However, operating at high temperatures often entails additional energy costs. In our pursuit of a more environmentally friendly process, we also examined the efficiency of PET degradation using a two-enzyme system at room temperature (30°C). Remarkable improvements in achieving complete TPA conversion were also observed at room temperature (**Fig. S33, Text S9**), emphasizing the potential of coupling BHETase into PET degradation as an alternative method for effectively degrading plastics without needing high temperatures. Furthermore, the BHETases successfully address the challenge of BHET/MHET accumulation during TPA production, which can result in the loss of mechanical properties in the resynthesized PET. This significant achievement reinforces our conviction that the newly discovered BHETase plays a crucial role in the puzzle of PET biodegradation patterns.

To clarify these points, we included the following sentences in the Revised manuscript: *Page 5, line 91-97:*

Furthermore, we utilized the engineered Δ BHETases (Δ BsEst and Δ ChryBHETase) in a two-enzyme system to generate homogeneous TPA through PET biodegradation. We demonstrated the feasibility of coupling Δ BHETases with all seven state-of-the-art PET hydrolases, namely PETase¹, DepoPETase⁴, FAST-PETase⁶, DuraPETase², ThermoPETase³, LCC⁷ and LCC-ICCG⁵.

Pages 12, line 234-240:

As indicated in **Table S9**, both BHETase and Δ BHETase maintained substantial activity within the temperature range of 50 °C to 60 °C. Notably, the engineered Δ BHETase variant Δ BsEst demonstrated superior thermostability compared to the wild type, with a remarkable half-life of up to 38.23 h at 60 °C. These findings underscore the potential of Δ BHETase in sustaining activity and stability even at elevated temperatures, positioning it as an up-and-coming candidate for PET degradation applications.

Pages S64, line 958-962 (SI):

Table S1 Half-life of BHETases and Δ BHETase at different temperatures.

	50 °C	55 °C	60 °C	65 °C	70 °C
BsEst	89.64±4.67 h	62.36±2.99 h	33.07±2.87 h	7.88±0.65 h	0.65±0.11 h
Δ BsEst	90.08±5.82 h	60.36±4.51 h	38.23±2.09 h	9.57±1.02 h	0.97±0.23 h
ChryBHETase	1.81±0.23 h	0.40±0.09 h	0.17±0.04 h	0.10±0.01 h	0.04±0.01 h
Δ ChryBHETase	1.92±0.67 h	0.66±0.35 h	0.18±0.03 h	0.09±0.02 h	0.05±0.01 h

Page 18-22, line 324-393:

2.4 Full TPA conversion from PET achieved by Δ BHETase in a simple two-enzyme degradation system

In 2016, Yoshida et al. published a groundbreaking study introducing *Is*PETase¹. This enzyme exhibited superior activity to Tfh, LCC, and FsC when degrading commercial bottle-derived PET and PET film under ambient conditions. This discovery shed light on the potential to mitigate the environmental impact of uncollectable PET release, offering a promising avenue beyond current recycling methods^{11–14}. Nevertheless, TPA produced by PETase often suffered from contamination by oligo ethylene terephthalates, BHET, and MHET, which posed limitations on downstream applications^{15–17}. To address this challenge, we have developed a two-enzyme degradation system that combines different PET hydrolases, including PETase¹, DepoPETase⁴, FAST-PETase⁶, DuraPETase², ThermoPETase³, LCC⁷ and LCC-ICCG⁵, with Δ BHETase. This innovative system enables the production of pure TPA, as illustrated in **Fig. 5a**.

To investigate the viability of Δ BHETase at high temperatures, we conducted experiments under a representative reaction condition of 60 °C. This choice of temperature aligns with commonly employed conditions in various PET degradation studies^{4–6,8}. As depicted in **Fig. 5b–h**, coupling Δ BHETase with all seven PET hydrolases in a two-enzyme system resulted in a substantial increase in TPA production compared to using PET hydrolase alone in a single-enzyme system. Interestingly, our study did not observe complete hydrolysis of PET films into homogeneous TPA within 120 h for any single PET hydrolases under the current depolymerization conditions (as shown in **Fig. 5b–h**). Instead, significant amounts of BHET+MHET were detected during degradation in almost all single-enzyme systems (**Fig. 5i**). These results differ from previous studies on single FAST-PETase⁶, and LCC-ICCG⁵. Unfavorable circumstances and/or the crystallinity of the plastics may be contributing factors to this outcome^{9,10}. In contrast, the incorporation of Δ BHETases into a single depolymerization system at 60 °C resulted in the complete hydrolysis of PET films to TPA, as observed in DepoPETase- Δ BHETase and FAST-PETase- Δ BHETase (**Fig. 5c–d** and **Fig. S31**). Specifically, when combined with DepoPETase, Δ BsEst demonstrated a 1.8-fold TPA yield increase compared to DepoPETase alone. Similarly, in combination with FAST-PETase, Δ ChryBHETase exhibited a 1.6-fold increase, and Δ BsEst achieved an impressive 2.0-fold increase compared to FAST-PETase alone. These notable enhancements can be attributed to the Δ BHETases' higher affinity for the intermediate product BHET and their ability to maintain the superior turnover efficiency than all state-of-the-art PET hydrolase (**Fig. S32**, TPA conversion rate of Δ BsEst is 31.6 mM/h/mg_{enzyme}, and Δ ChryBHETase is 14.4 mM/h/mg_{enzyme}). Intriguingly, even without additional enzymes or components, purified Δ BsEst and Δ ChryBHETase

(data not shown) generated trace amounts of MHET and TPA from the PET film. This discovery underscores the remarkable capability of Δ BHETase to directly degrade PET, serving as a pivotal catalyst in the process.

The discovery as mentioned earlier was further supported by the analysis of the PET film surface using visible photography, scanning electron microscopy (SEM), and the measurement of reduced water contact angle (**Fig. 5j**). Specifically, the two-enzyme system exhibited a remarkable ability to effectively degrade the PET film with up to 22° water contact angle decrease, resulting in a visibly rougher surface compared to the single enzyme system. To address the additional energy costs associated with high temperatures, we also investigated the PET degradation efficiency of the two-enzyme system at room temperature (30 °C) to achieve a more environmentally friendly process. And we observed similar achievements at room temperature compared to at 60 °C (**Fig. S33**). Remarkably, PETase/ Δ BsEst and PETase/ Δ ChryBHETase produced 663.1 μ M (7.0-fold) and 617.6 μ M (6.5-fold) of TPA within 24 h, far more than PETase alone (95 μ M). Summarily, these significant achievements reinforce our conviction that the newly discovered BHETase plays a crucial role in the puzzle of PET biodegradation patterns. Further details regarding the two-enzyme system can be found in **Text 9** and **Fig. S33** in **SI**.

Fig. 5 Two-enzyme degradation system at 60 °C. (a) The overview of the two-enzyme degradation system. Time course of PET depolymerization in a two-enzyme system with different PET hydrolases, including (b) PETase, (c) DepoPETase, (d) FAST-PETase, (e) DuraPETase, (f) ThermoPETase, (g) LCC, and (h) LCC-ICCG. (i) The PET monomers (the sum of BHET and MHET) released at 96 h. (j) The SEM images (up panel), and water contact angle analysis (down panel) of the PET film in a single-enzyme degradation system, two-enzyme degradation system with $\Delta BsEst$, and two-enzyme degradation system with $\Delta ChryBHETase$ after 48 h at 60 °C. Reaction condition: The PET films ($\phi=6$ mm) were soaked in 2940 μ L of $Na_2HPO_4-NaH_2PO_4$ (pH 8.5, 50 mM) buffer at 60 °C with 50 μ L of 0.5 mg/mL PET hydrolases and $\Delta BHETases$, respectively. The fresh enzyme solution was supplemented every 24 h for depolymerization. Error bars represent the standard deviation of three replicate experiments.

Pages S9-10, line 305-351 (SI):

In Text 9:

After 96 h at 60 °C, the concentrations of products increased slowly with time. We speculated that the product inhibition might be the reason for this phenomenon. Notably, at 96 h, for PET hydrolases with great thermostability, including DepoPETase, FAST-PETase and LCC-ICCG, the total amount of product released by a two-enzyme system with $\Delta BHETases$ was up to 1.3-fold compared to the single-enzyme system; For PET hydrolases with poor thermostability, such as PETase, LCC, DuraPETase and ThermoPETase, the total amount of products released by a two enzyme system with $\Delta BHETases$ was up to 1.6-fold compared to the single-enzyme system (**Fig. 5b-h**). This synergy effect was more significant in the case of degrading PET film. The findings in **Fig. 5j** provide visual and microscopic evidence of the enhanced PET degradation achieved by implementing the two-enzyme system, further validating its efficacy and potential for practical applications.

In previous studies^{2,4-6,8}, researchers have commonly opted for high temperatures to exploit the ability of heat to reduce the crystallinity of PET. This decrease in crystallinity promotes plastic decomposition, rendering it more amenable to degradation processes. However, operating at elevated temperatures often incurs additional energy costs. In our endeavor to develop a more environmentally friendly process, we investigated the efficiency of PET degradation using a two-enzyme system at room temperature (30 °C). As shown in **Fig. S33a**, involving BHETase and $\Delta BHETase$ in PET degradation enabled to offset of the poor ability of PETase in converting BHET to TPA. Notably, PETase+ $\Delta BsEst$ can achieve a 100% yield of TPA in 4 h, but PETase+ $BsEst$ with 94.8%. Such synergy effect was more significant in the

case of degrading PET film (**Fig. S33b**). TPA is obtained through the action of (i) PETase alone, (ii) PETase+B_sEst, (iii) PETase+ΔB_sEst, (iv) PETase+ChryBHETase, and (v) PETase+ΔChryBHETase. **Notably, PETase+ΔBHETase had up to 7-fold improvement in TPA yield (e.g., ΔB_sEst: 663.1 μM) compared to single PETase (95 μM).** In contrast, with PETase, the amounts of BHET, MHET, and TPA were measured reliably and satisfactorily, determined to be 29.3 μM, 463.3 μM, and 95 μM within 24 h, respectively. As depicted in **Fig. S33b**, the two-enzyme approach successfully reduced the concentration of MHET to undetectable levels. Furthermore, the highest TPA concentrations were observed with PETase+ΔB_sEst, reaching 663.1 μM, compared to PETase+B_sEst (617.6 μM of TPA). Similarly, PETase+ChryBHETase and PETase+ΔChryBHETase exhibited enhanced TPA production, with measured concentrations of 561.3 μM and 589.6 μM, respectively, surpassing the anticipated levels. These findings demonstrate the effectiveness of the two-enzyme system in achieving higher TPA yields compared to single-enzyme systems and underline the potential of ΔBHETases in enhancing PET degradation and TPA production. **This founding was also proved by analyzing the surface of the PET film shown by visible photography, scanning electron microscopy (SEM), and the reduced water contact angle (Fig. S33c).** Although the short detection time (24 h) slightly hid the outstanding properties of ΔBHETases (e.g., 5-7.3% improved conversion compared to BHETase), PETase+ΔBHETases enabled to degrade PET film strongly resulting in much rougher surfaces (**Fig.S33c**, middle panel). Overall, the simple two-enzyme degradation system, especially with ΔBHETase, can efficiently produce pure TPA.

Page S53, line 896-904 (SI):

Fig. S1 Two-enzyme degradation system consisting of PETase and Δ BHETase. The biodegradation performance of PETase and two-enzyme system towards **(a)** BHET and **(b)** PET film. Reaction condition: 5 mM BHET or PET films ($\phi=6$ mm) were soaked in 2940 μ L of $\text{Na}_2\text{HPO}_4\text{-NaH}_2\text{PO}_4$ (pH 8.5, 50 mM) buffer at 30 °C with 50 μ L of 0.5 mg/mL PET hydrolases and Δ BHETases for 24 h, respectively. Error bars represent the standard deviation of three replicate experiments. **(c)** The visual photography (up panel) and SEM images (middle panel), and water contact angle analysis of the PET film after biodegradation.

Comment 2:

*While the authors showcase production of *p*-phthaloyl chloride as an exemplar product (likely due to its appearance in products like Kevlar), this will (1) not solve the plastics issue as the demand for this molecule is far less than the amount of PET produced and (2) this does not create a circular economy with respect to plastics / PET.*

Addressed and included.

This is an insightful point. As suggested by reviewer 1, we developed a fully closed-loop PET recycling process utilizing Δ BHETases to produce the pure monomer TPA, followed by resynthesizing new PET (**Section 2.5, Fig. 6, Fig. S37**). **We fully agree** that only upcycling the TPA into *p*-phthaloyl chloride cannot meet the PET market demand in society. In **Section 2.5, we successfully resynthesized** the produced TPA monomer obtained from chemical-enzymatic catalysis into virgin PET, achieving up to 554 mg out of 920 mg TPA. The latter accomplished a complete cycle of PET from degradation to repolymerization. Instead of emphasizing the upcycling of plastic waste into a single exemplary chemical *p*-phthaloyl chloride, this PET recycle results were included in the revised manuscript (**More details were transferred to SI**) to highlight the potential of our method for contributing to the more sustainable and circular plastic economy.

Page 2, line 32-37:

Finally, we developed a Δ BHETase-joined tandem chemical-enzymatic approach to valorize 21 commercial post-consumed plastics into virgin PET and an exemplary chemical (*p*-phthaloyl chloride) for achieving the closed-loop PET recycling and open-loop PET upcycling.

Page 5, line 97-103:

As biotechnological degradation and recycling of plastics are still in the early stages of the learning curve, we further developed a tandem chemical and biological PET degradation system that leverages the advantages of both depolymerization processes to expand the applicability of BHETases fully. To achieve fully closed-loop PET recycling and open-loop PET upcycling, respectively, the obtained pure TPA was used

for resynthesizing recycled PET and valorizing TPA into the valuable chemical product (*p*-phthaloyl chloride as an exemplary chemical).

Page 22-24, line 394-445:

2.5 The closed-loop recycling and open-loop PET upcycling based on the BHETase-joined tandem chemical and biological PET degradation system

Although our results and similar works confirmed that biotechnological degradation has unprecedented potential for PET recycling^{18,19}, enzymatic hydrolysis often has low substrate loads and slower PET degradation compared to certain thermochemical methods¹⁹⁻²². To fully explore the potential of applying $\Delta BsEst$ and $\Delta ChryBHETase$ into practical PET re/upcycling, we developed a tandem chemical-biological approach to leverage the advantages of chemical and biological depolymerization processes. This approach involves a chemical pre-treatment method called glycolysis at the first stage to efficiently degrade PET into low molecular weight hydrolysate BHET within a short time frame while minimizing excessive decomposition^{23,24}. Subsequently, $\Delta BHETases$ contributed to yield pure TPA from the intermediate BHET.

As shown in **Fig. 6a**, K_2CO_3 as a chemical catalyst was selected due to its biocompatibility compared to the most active glycolysis catalysts but toxic Zn salts²⁵. Twenty-one post-consumer plastic products were collected for practical examination, including beverage packaging, food packaging, and household plastic fields (**Table S12**). To evaluate our tandem degradation strategy, a large sample pool of 21 commercial post-consumed plastic waste with crystallinity ranging from 1.54% to 38.73% was used (**Table S12**). Significantly, the BHET yield obtained from PET glycolysis reached an impressive level of over 81.5% (**Fig. S34a and Fig. S35**). Moreover, after undergoing filtration and recrystallization processes, the purity of the BHET achieved was above 95.5% (**Fig. S34a and Fig. S35**). Moving forward, the $\Delta BHETase$ hydrolysis step resulted in the production of TPA with up to 84.1% yield (**Fig. S34b**). These findings demonstrate the efficiency and effectiveness of the combined PET glycolysis and $\Delta BHETase$ hydrolysis processes for various plastic waste degradation and TPA production.

Motivated by the previous results and to complete a closed-loop PET recycling, we resynthesized 554 mg virgin PET directly from the chemical-enzymatic degradation solution (920 mg of TPA) (**Fig. 6a and Fig.S36**). And the entire cycle from degradation to repolymerization can be completed within a few days (**Fig. 6b and S37a**). These results confirmed the viability of producing virgin PET from non-petroleum sources using our tandem PET recycling system in a closed-loop manner. Besides, converting

plastic waste into various high-value chemicals (e.g., fuels, waxes, lubricants) offers an open-loop upcycling perspective to address the urgent issues of plastic waste. Herein, *p*-phthaloyl chloride, as a single exemplary chemical, was synthesized using the TPA yielding from plastic wastes to achieve an open-loop PET upcycling (Fig. 6c and Fig. S37b, see more details in SI Text 10). Collectively, leveraging the engineered Δ BHETase, we have successfully demonstrated a process concept that BHETase-joined tandem PET degradation system to transform plastic waste into valuable products. These innovative technologies hold tremendous potential in supporting a circular plastics economy, where plastic waste is efficiently recycled and repurposed.

Fig. 6 | The closed-loop recycling and open-loop PET upcycling based on the BHETase-joined tandem chemical and biological PET degradation system. (a) The schematic of the closed-loop PET recycling and open-loop PET upcycling pathway. (b) The schematic of the closed-loop PET recycling. (c) The schematic of the open-loop PET upcycling. (d) Theoretical maximum synthesis yield of *p*-phthaloyl chloride from

21 commercial pc-products. Chemical glycolysis conditions: glycolysis of 5 g PET, 3 h of reaction time, 200 °C of temperature, and 0.1 g of K₂CO₃ loading. Enzymatic catalysis condition: 2ml of 0.5 mg/ml Δ BsEst and Δ ChryBHETase hydrolyzing the BHET obtained from PET glycolysis at 30 °C within 24 h, respectively. Error bars represent the standard deviation of three replicate experiments.

Page S55, line 918-923 (SI):

Fig. S35 Liquid chromatography-mass spectrometry (LC-MS) data of BHET, TPA, and DMT. (a) LC-MS data of BHET by chemical glycolysis; (b) LC-MS data of TPA after enzymatic catalysis by Δ BHETase; (c) LC-MS data of DMT synthesized from TPA. (d) LC-MS data of p-phthaloyl chloride synthesized from TPA.

Fig. S36 DSC trace of virgin PET regenerated from the degraded solutions. The crystallinity of this regenerated PET is 45.78%. The melting onset is 210.49 °C. The melting peak temperature is 255.99 °C.

Comment 3:

Error bars are never really described here as to their meaning and replicates.

Addressed.

Thank you very much for pointing it out. The meaning and replicates of error bars at each figure have been described.

line 114:

All measurements were conducted in triplicate ($n = 3$), and the mean values were used for generating the heat maps.

line 174, 242, 324, 365 and supplementary line 667,674, 680, 816:

Error bars represent the standard deviation of three replicate experiments.

line 182 and supplementary line 775, 795:

Data plotted from the average of three independent MD runs.

line 294 and supplementary line 733, 740, 770:

Error bars showed the standard deviation from three independent MD runs for each simulation.

References

1. Yoshida, S. *et al.* A bacterium that degrades and assimilates poly(ethylene terephthalate). *Science* **351**, 1196–1199 (2016).
2. Cui, Y. *et al.* Computational redesign of a PETase for plastic biodegradation under ambient condition by the GRAPE strategy. *ACS Catal.* **11**, 1340–1350 (2021).

-
3. Son, H. F. *et al.* Rational protein engineering of thermo-stable PETase from *Ideonella sakaiensis* for highly efficient PET degradation. *ACS Catal.* **9**, 3519–3526 (2019).
 4. Shi, L. *et al.* Complete depolymerization of PET waste by an evolved PET hydrolase from directed evolution. *Angew. Chem. Int. Ed.* **62**, 1–11 (2023).
 5. Tournier, V. *et al.* An engineered PET depolymerase to break down and recycle plastic bottles. *Nature* **580**, 216–219 (2020).
 6. Lu, H. *et al.* Machine learning-aided engineering of hydrolases for PET depolymerization. *Nature* **604**, 662–667 (2022).
 7. Sulaiman, S. *et al.* Isolation of a novel cutinase homolog with polyethylene terephthalate-degrading activity from leaf-branch compost by using a metagenomic approach. *Appl. Environ. Microbiol.* **78**, 1556–1562 (2012).
 8. Bell, E. L. *et al.* Directed evolution of an efficient and thermostable PET depolymerase. *Nat. Catal.* **5**, 673–681 (2022).
 9. Serghei, A., Tress, M. & Kremer, F. The glass transition of thin polymer films in relation to the interfacial dynamics. *J. Chem. Phys.* **131**, 154904 (2009).
 10. Xia, W., Hsu, D. D. & Keten, S. Molecular Weight Effects on the Glass Transition and Confinement Behavior of Polymer Thin Films. *Macromol. Rapid Commun.* **36**, 1422–1427 (2015).
 11. Gewert, B., Plassmann, M. M. & MacLeod, M. Pathways for degradation of plastic polymers floating in the marine environment. *Environ. Sci. Process. Impacts* **17**, 1513–1521 (2015).
 12. Satti, S. M., Shah, A. A., Auras, R. & Marsh, T. L. Isolation and characterization of bacteria capable of degrading poly(lactic acid) at ambient temperature. *Polym Degrad Stab* **144**, 392–400 (2017).
 13. Auta, H. S., Emenike, C. U. & Fauziah, S. H. Screening of *Bacillus* strains isolated from mangrove ecosystems in Peninsular Malaysia for microplastic degradation. *Env. Pollut* **231**, 1552–1559 (2017).
 14. Jia, X., Qin, C., Friedberger, T., Guan, Z. & Huang, Z. Efficient and selective degradation of polyethylenes into liquid fuels and waxes under mild conditions. *Sci. Adv.* **2**, e1501591 (2016).
 15. Barth, M. *et al.* A dual enzyme system composed of a polyester hydrolase and a carboxylesterase enhances the biocatalytic degradation of polyethylene terephthalate films. *Biotechnol. J.* **11**, 1082–1087 (2016).
 16. Mrigwani, A., Thakur, B. & Guptasarma, P. Conversion of polyethylene terephthalate into pure terephthalic acid through synergy between a solid-degrading cutinase and a reaction intermediate-hydrolysing carboxylesterase. *Green Chem.* **24**, 6707–6719 (2022).
 17. Knott, B. C. *et al.* Characterization and engineering of a two-enzyme system for plastics depolymerization. *Proc. Natl. Acad. Sci.* **117**, 25476–25485 (2020).
 18. Wei, R. *et al.* Possibilities and limitations of biotechnological plastic degradation and recycling. *Nat. Catal.* **3**, 867–871 (2020).
 19. Kim, H. T. *et al.* Chemo-Biological Upcycling of Poly(ethylene terephthalate) to Multifunctional Coating Materials. *ChemSusChem.* **14**, 4251–4259 (2021).

-
20. Palm, G. J. *et al.* Structure of the plastic-degrading *Ideonella sakaiensis* MHETase bound to a substrate. *Nat Commun* **10**, 1717 (2019).
 21. Knott, B. C. *et al.* Characterization and engineering of a two-enzyme system for plastics depolymerization. *Proc. Natl. Acad. Sci.* **117**, 25476–25485 (2020).
 22. Barragán-Iglesias, P. *et al.* Inhibition of Poly(A)-binding protein with a synthetic RNA mimic reduces pain sensitization in mice. *Nat. Commun.* **9**, 10 (2018).
 23. Castro, A. M. de & Carniel, A. A novel process for poly(ethylene terephthalate) depolymerization via enzyme-catalyzed glycolysis. *Biochem. Eng. J.* **124**, 64–68 (2017).
 24. de Castro, A. M., Carniel, A., Nicomedes Junior, J., da Conceição Gomes, A. & Valoni, É. Screening of commercial enzymes for poly(ethylene terephthalate) (PET) hydrolysis and synergy studies on different substrate sources. *J. Ind. Microbiol. Biotechnol.* **44**, 835–844 (2017).
 25. López-Fonseca, R., Duque-Ingunza, I., de Rivas, B., Arnaiz, S. & Gutiérrez-Ortiz, J. I. Chemical recycling of post-consumer PET wastes by glycolysis in the presence of metal salts. *Polym Degrad Stab* **95**, 1022–1028 (2010).

Reviewer #2:

General Comments to the Authors: *The paper provides some new information on BHET hydrolyzing enzymes. Despite these novel insights, the paper lacks a solid comparison with the many existing PET hydrolyzing enzyme systems (identical reactions conditions, substrates etc).*

We greatly thank Reviewer 2 for the comment “*The paper provides some new information on BHET hydrolyzing enzymes.*” and for giving valuable comments to help us improve the manuscript. According to Reviewer 2’s comments, we utilized the engineered Δ BHETases (Δ BsEst and Δ ChryBHETase) in a two-enzyme system to successfully generate homogeneous TPA through PET biodegradation. We demonstrated the feasibility of coupling Δ BHETases with all seven state-of-the-art PET hydrolases, namely PETase¹, DepoPETase², FAST-PETase³, DuraPETase⁴, ThermoPETase⁵, LCC⁶ and LCC-ICCG⁷. Detailed results were described in **section 2.4** and **Text 9** in **SI** (see below). As depicted in **Fig 5c-i**, the coupling of Δ BHETase with all seven PET hydrolases in a two-enzyme system resulted in a substantial increase in TPA production compared to using PET hydrolase alone in a single-enzyme system. At the initial 48 h, we found that the assistance of Δ BHETases improved the TPA production rate, achieving 1.3-fold (Δ ChryBHETase+DepoPETase) and 1.8-fold (Δ BsEst+DepoPETase) compared to DepoPETase; 1.6-fold (Δ ChryBHETase+FAST-PETase) and 2.0-fold (Δ BsEst+FAST-PETase) compared to FAST-PETase. After 96 h, for PET hydrolases with great thermostability, including DepoPETase, FAST-PETase, and LCC-ICCG, the total amount of product released by a two-enzyme system with Δ BHETases was up to 1.3-fold compared to the single-enzyme system. Whereas for those with poor thermostability, such as PETase, LCC, DuraPETase and ThermoPETase, the total amount of products released by a two-enzyme system with Δ BHETases was up to 1.6-fold compared to the single-enzyme system.

To clarify these points, we included the following sentences in the Revised manuscript:

Page 18-22, line 324-393:

2.4 Full TPA conversion from PET achieved by Δ BHETase in a simple two-enzyme degradation system

In 2016, Yoshida et al. published a groundbreaking study introducing *Is*PETase¹. This enzyme exhibited superior activity to TfH, LCC, and FsC when degrading commercial bottle-derived PET and PET film under ambient conditions. This discovery shed light

on the potential to mitigate the environmental impact of uncollectable PET release, offering a promising avenue beyond current recycling methods⁸⁻¹¹. Nevertheless, TPA produced by PETase often suffered from contamination by oligo ethylene terephthalates, BHET, and MHET, which posed limitations on downstream applications¹²⁻¹⁴. To address this challenge, we have developed a two-enzyme degradation system that combines different PET hydrolases, including PETase¹, DepoPETase², FAST-PETase³, DuraPETase⁴, ThermoPETase⁵, LCC⁶ and LCC-ICCG⁷, with Δ BHETase. This innovative system enables the production of pure TPA, as illustrated in **Fig. 5a**.

To investigate the viability of Δ BHETase at high temperatures, we conducted experiments under a representative reaction condition of 60 °C. This choice of temperature aligns with commonly employed conditions in various PET degradation studies^{2,3,7,15}. As depicted in **Fig. 5b-h**, coupling Δ BHETase with all seven PET hydrolases in a two-enzyme system resulted in a substantial increase in TPA production compared to using PET hydrolase alone in a single-enzyme system. Interestingly, our study did not observe complete hydrolysis of PET films into homogeneous TPA within 120 h for any single PET hydrolases under the current depolymerization conditions (as shown in **Fig. 5b-h**). Instead, significant amounts of BHET+MHET were detected during degradation in almost all single-enzyme systems (**Fig. 5i**). These results differ from previous studies on single FAST-PETase³, and LCC-ICCG⁷. Unfavorable circumstances and/or the crystallinity of the plastics may be contributing factors to this outcome^{16,17}. In contrast, the incorporation of Δ BHETases into a single depolymerization system at 60 °C resulted in the complete hydrolysis of PET films to TPA, as observed in DepoPETase- Δ BHETase and FAST-PETase- Δ BHETase (**Fig. 5c-d** and **Fig. S31**). Specifically, when combined with DepoPETase, Δ BsEst demonstrated a 1.8-fold TPA yield increase compared to DepoPETase alone. Similarly, in combination with FAST-PETase, Δ ChryBHETase exhibited a 1.6-fold increase, and Δ BsEst achieved an impressive 2.0-fold increase compared to FAST-PETase alone. These notable enhancements can be attributed to the Δ BHETases' higher affinity for the intermediate product BHET and their ability to maintain the superior turnover efficiency than all state-of-the-art PET hydrolase (**Fig. S32**, TPA conversion rate of Δ BsEst is 31.6 mM/h/mg_{enzyme}, and Δ ChryBHETase is 14.4 mM/h/mg_{enzyme}). Intriguingly, even without additional enzymes or components, purified Δ BsEst and Δ ChryBHETase (data not shown) generated trace amounts of MHET and TPA from the PET film. This discovery underscores the remarkable capability of Δ BHETase to directly degrade PET, serving as a pivotal catalyst in the process.

The discovery mentioned above was further supported by the analysis of the PET film surface using visible photography, scanning electron microscopy (SEM), and the

measurement of reduced water contact angle (**Fig. 5j**). Specifically, the two-enzyme system exhibited a remarkable ability to effectively degrade the PET film with up to 22° water contact angle decrease, resulting in a visibly rougher surface compared to the single enzyme system. To address the additional energy costs associated with high temperatures, we also investigated the PET degradation efficiency of the two-enzyme system at room temperature (30 °C) to achieve a more environmentally friendly process. And we observed similar achievements at room temperature compared to at 60 °C (**Fig. S33**). Remarkably, PETase/ $\Delta BsEst$ and PETase/ $\Delta ChryBHETase$ produced 663.1 μM (7.0-fold) and 617.6 μM (6.5-fold) of TPA within 24 h, far more than PETase alone (95 μM). Summarily, these significant achievements reinforce our conviction that the newly discovered BHETase plays a crucial role in the puzzle of PET biodegradation patterns. Further details regarding the two-enzyme system can be found in **Text 9** and **Fig. S33** in **SI**.

Fig. 5 Two-enzyme degradation system at 60 °C. (a) The overview of the two-enzyme degradation system. Time course of PET depolymerization in a two-enzyme system with different PET hydrolases, including (b) PETase, (c) DepoPETase, (d) FAST-PETase, (e) DuraPETase, (f) ThermoPETase, (g) LCC, and (h) LCC-ICCG. (i) The PET monomers (the sum of BHET and MHET) released at 96 h. (j) The SEM images (up panel), and water contact angle analysis (down panel) of the PET film in a single-enzyme degradation system, two-enzyme degradation system with $\Delta BsEst$, and two-enzyme degradation system with $\Delta ChryBHETase$ after 48 h at 60 °C. Reaction condition: The PET films ($\phi=6$ mm) were soaked in 2940 μL of $\text{Na}_2\text{HPO}_4\text{-NaH}_2\text{PO}_4$ (pH 8.5, 50 mM) buffer at 60 °C with 50 μL of 0.5 mg/mL PET hydrolases and $\Delta BHETases$, respectively. The fresh enzyme solution was supplemented every 24 h for depolymerization. Error bars represent the standard deviation of three replicate experiments.

Comment 1:

Why was diethyl terephthalate used?

Explained and included.

Diethyl terephthalate (DET) is commonly used for PET degradation microorganisms screening because it is a small molecule structurally similar to polyethylene terephthalate (PET), the plastic PET degradation microorganisms degrade. DET can be easily hydrolyzed by PET hydrolases, releasing terephthalic acid and ethanol that can be detected by various analytical methods¹⁸. DET as a screening tool can quickly identify and isolate microorganisms that produce PET hydrolases with high activity and specificity towards PET^{19,20}. Overall, using DET for PET degradation microorganisms screening is a convenient and effective method for identifying microorganisms with the potential to degrade PET²¹.

To clarify the points mentioned above, we included the following sentences in the Revised Manuscript:

Page 6, line 113-117:

The common substrate Diethyl terephthalate (DET) is used for enriching PET degradation microorganisms because DET is a small molecule structurally similar to PET¹⁸. DET can be easily hydrolyzed by PET degradation microorganisms, releasing terephthalic acid and ethanol, which can be detected by various analytical methods like high-performance liquid chromatography (HPLC)¹⁹⁻²¹.

Comment 2:

L 117 onwards, very difficult to read, maybe put all references to figures at the end of sentences

Addressed. The references to figures were moved to the end of the sentences.

Page 7, line 132-143:

In Phase III, sequence identity analysis and *in vitro* characterization of the purified *BsEst* and *ChryBHETase* with degrading PET powder, BHET, and *pNP*-aliphatic esters confirmed that *ChryBHETase* prefers BHET to aliphatic esters (**Fig. 1d-e, Fig. 2c-e and Fig. S5-7**), compared with lipase (BSLA, Identity of *ChryBHETase* vs. BSLA is 13.18%), esterase (*Bs2Est*, 25.55%; *TfCa*, 24.66%), PETase (12.41%) and MHETase^{1,22} (14.31%, **Fig. 2c and Fig. S8-10**), leading to its designation as the BHET hydrolase (termed BHETases).

Comment 3:

L 124 please rewrite, all PET/BHET hydrolyzing enzymes are esterase, is there a new EC enzyme number for BHETases?

Addressed.

Page 7, line 139:

“*BsEst*, in principle, belonged to esterase regarding the sequence identify analysis (**Fig. S9c**), but its preferred BHETase activity encourages us to treat it as a BHETase” changed to “*BsEst* belonged to one kind of esterases regarding the sequence identify analysis (**Fig. S9c**), but its preferred BHETase activity encourages us to treat it as a BHETase”.

Currently, there is no new EC enzyme number specifically designated for BHETases. Using the term BHETase distinguishes them from other esterases mainly based on their ability to degrade the specific BHET.

Comment 4:

Legend Fig 2e, upon BHET -> upon incubation on BHET
Done.

Thanks for bringing this to our attention. “upon BHET” changed to “**upon incubation on BHET**”.

Comment 5:

Other longer-chain oligomers should be assessed as substrates for the enzyme.

Addressed and included.

We fully agree that more tests would help assess the substrate specificity of two BHETases (*BsEst* and *ChryBHETase*). Therefore, we have performed substrate specificity experiments on *pNP* ester substrates of different carbon chain lengths,

phenyl esters, glycerol esters and tertiary alcohols, and the relevant content and data have been added to the **Supplementary information**.

Page S6, line 210-214 (SI):

Additionally, the substrate specificity of BHETases for other phenyl esters, glycerides, tertiary alcohols was also investigated in **Fig.S7b**. Results indicated that *BsEst* showed significant hydrolysis of almost all substrates tested in this study, while *ChryBHETase* showed significant hydrolysis of 2-methylbutyl acetate, glyceryl tributyrates and olive oil substrates.

Page S27, line 733-742 (SI):

Fig. S7 The substrate specificity of *BsEst* and *ChryBHETase*. (a) Specific activity of *BsEst* and *ChryBHETase* with different carbon chain lengths of *p*NPB (C₄), *p*NPC (C₈), and *p*NPP (C₁₂) as substrate. Reaction condition: the crude enzymes were incubated with *p*NPB, *p*NPC and *p*NPP in a buffer containing 50 mM Na₂HPO₄-NaH₂PO₄ (pH 7.5) at 40 °C. Error bars represent the standard deviation of three replicate experiments. (b) Substrate specificity of *BsEst* and *ChryBHETase* for other phenyl esters, glycerides, tertiary alcohols. Abbreviation: GTO: Glyceryl trioctanoate; PA: Phenyl acetate; 2-MA: 2-Methylbutyl acetate; LA: Linalyl acetate; GTB: Glyceryl tributyrates; OO: Olive oil. The deeper the yellow, the greater the hydrolysis capacity.

Comment 6:

L 199 Discuss kinetic parameters, 100% conversion after a certain does not tell much **Addressed**. We emphasized the kinetic parameters of BHETases and Δ BHETases as follows:

Page 12, line 227-231:

Both Δ *ChryBHETase* and Δ *BsEst* showed significantly increased turnover numbers (*k_{cat}*, Δ *ChryBHETase*: 2.6-fold, Δ *BsEst*: 1.7-fold) compared to *ChryBHETase* and

BsEst. Reasonably, catalytic efficiency (k_{cat}/K_M) of $\Delta BsEst$ and $\Delta ChryBHETase$ had 3.5-fold and 1.5-fold enhancement, respectively.

Comment 7:

L 212 provides half-life times at different temperatures rather than percentual values for arbitrary time points

Addressed. According to the reviewer's comments, the experiment of half-life times at different temperatures has been supplemented. The corresponding experimental details, experimental data, and discussion were added to the main text and the Supplementary Information.

Page 12, line 234-240:

As indicated in **Table S9**, both BHETase and $\Delta BHETase$ maintained substantial activity within the temperature range of 50 °C to 60 °C. Notably, the engineered $\Delta BHETase$ variant $\Delta BsEst$ demonstrated superior thermostability compared to the wild type, with a remarkable half-life of up to 38.23 h at 60 °C. These findings underscore the potential of $\Delta BHETase$ in sustaining activity and stability even at elevated temperatures, positioning it as an up-and-coming candidate for PET degradation applications.

Page 28, line 565-569:

The purified proteins were stored separately at 50-70 °C (at 5°C intervals) and the remaining enzyme activity was tested every 30 min until it was reduced to half of the initial enzyme activity. Based on the enzyme inactivation equation: $[E_t] = [E_0] \exp(-kt)$, E_t : relative enzyme activities at time t; E_0 : initial relative enzyme activities. Thus, half-life $t_{1/2} = \ln 2/k$.

Pages S64, line 958-962 (SI):

Table S2 Half-life of BHETases and $\Delta BHETase$ at different temperatures.

	50 °C	55 °C	60 °C	65 °C	70 °C
BsEst	89.64±4.67 h	62.36±2.99 h	33.07±2.87 h	7.88±0.65 h	0.65±0.11 h
$\Delta BsEst$	90.08±5.82 h	60.36±4.51 h	38.23±2.09 h	9.57±1.02 h	0.97±0.23 h
ChryBHETase	1.81±0.23 h	0.40±0.09 h	0.17±0.04 h	0.10±0.01 h	0.04±0.01 h
$\Delta ChryBHETase$	1.92±0.67 h	0.66±0.35 h	0.18±0.03 h	0.09±0.02 h	0.05±0.01 h

Comment 8:

l294 what does highest activity towards amorphous PET mean? There are enzymes that are much more efficient, provide clear numbers for such statements

Explained and included. We intended to mention that released compounds by PETase under mild conditions (30 °C) are far exceeded than TtH, LCC, and FsC, implying that

the activity of PETase was higher than Tfh, LCC, and FsC¹. These data were shown in **Fig. R1** as below in *Science*, 2016, Vol 351, Issue 6278¹).

Figure R1. The activity of the PET hydrolytic enzymes for highly crystallized PET (hcPET) and the effect of temperature on enzymatic PET film hydrolysis. Data obtained from Yoshida, et al. *Science*¹.

Page 18, line 326-331:

In 2016, Yoshida et al. published a groundbreaking study introducing *IsPETase*¹. This enzyme exhibited superior activity to Tfh, LCC, and FsC when degrading commercial bottle-derived PET and PET film under ambient conditions. This discovery shed light on the potential to mitigate the environmental impact of uncollectable PET release, offering a promising avenue beyond current recycling methods⁸⁻¹¹.

Comment 9:

l 303 the activity of the enzyme system should be compared in detail with the activity of other published polyester degrading enzymes such as the LCC at identical conditions. l 316 Efficient manner, - see above, a detailed comparison with the many other enzyme and enzyme systems published for PET hydrolysis is required.

Explained and addressed.

We fully agree that the data of the two-enzyme system at high temperature (60 °C) should be compared in detail with other state-of-the-art polyester degrading enzymes, such as PETase¹, DepoPETase², FAST-PETase³, DuraPETase⁴, ThermoPETase⁵, LCC⁶ and LCC-ICCG⁷. More information can be found in **section 2.4** and **Comment 1**.

Comment 10:

Fig 6b Plastic products hydrolysis, this would only be valuable when compared to other existing PET hydrolyzing enzymes and when detailed composition / properties (e.g. % crystallinity) of products would be given.

Addressed.

We fully agree that more detailed composition/properties of commercial post-consumed plastic would facilitate the completeness of the experimental data and the reader's access to accurate information. The corresponding experimental details and experimental data were added to the **Experimental Section 4.1.12** as well as the **Supplementary information (Fig. S12)**.

Page 23, line 410:

To evaluate our tandem degradation strategy, a large sample pool of 21 commercial post-consumed plastic waste with crystallinity ranging from 1.54% to 38.73% was used (**Table S12**).

Page 29, line 581-593:

4.1.12 Differential Scanning Calorimetry (DSC)

DSC was used to determine the percent crystallinity of the PET films collected from post-consumer(pc) PET products. The following protocol was used for each sample. A sample of 2-3 mg in an aluminum pan was cooled from room temperature to 0 °C. The pan was then heated from 0 to 300 °C at 10 °C min⁻¹ and then maintained at 300 °C for 1 min under a nitrogen atmosphere. Subsequently, the sample was quenched to 0 °C at 10 °C min⁻¹. The heat of fusion ΔH_m and cold crystallization ΔH_c were determined by integrating areas (J g⁻¹) under peaks. The percent crystallinity was calculated using the following equation:

$$\%crystallinity = \left[\frac{\Delta H_m - \Delta H_c}{\Delta H_m} \right] \times 100$$

where ΔH_m is the enthalpy of melting (J g⁻¹), ΔH_c is the enthalpy of cold crystallization (J g⁻¹), and ΔH_m is the enthalpy of melting for a 100% crystalline PET sample, which is 140.1 J g⁻¹ ^{4,7}.

Page 70, line 968-973 (SI):

Table S3 Twenty-one commercial post-consumed plastic products collected in this study ^a.

Category	Sample number	Post-consumer plastic products	Initial mass (g)	Crystallinity %
Beverage packaging	#1	Milk-tea container	5.00	6.39
	#2	COSTA container	5.00	12.30
	#3	Water container	5.00	2.70
	#4	Water container	5.00	4.40
	#5	Cola container	5.00	1.72
	#6	Coffee cap	5.00	2.89
	#7	Coffer container	5.00	10.76
Food	#8	Cookies container	5.00	5.37

packaging	#9	Sugar container	5.00	5.59	
	#10	Blueberry container	5.00	38.73	
	#11	Marinated plum container	5.00	0.07	
	#12	Lemon container	5.00	5.39	
	#13	Plum container	5.00	15.15	
	#14	Egg container	5.00	5.42	
	#15	Tomato container	5.00	7.23	
	#16	Pineapple container	5.00	5.03	
	#17	Pipette	5.00	3.31	
	Household goods packaging	#18	Hand sanitizer container	5.00	13.71
		#19	Repellent container	5.00	2.51
	Others	#20	Commercial PET film	5.00	4.46
		#21	Commercial PET powder	5.00	1.54

^a21 different post-consumed plastic products used in the packaging of food, beverages, medications, office supplies, household goods, and cosmetics available at local grocery store chains. The crystallinity % of the intact pc-PET films was determined by DSC.

Reference

1. Yoshida, S. *et al.* A bacterium that degrades and assimilates poly(ethylene terephthalate). *Science* **351**, 1196–1199 (2016).
2. Shi, L. *et al.* Complete depolymerization of PET waste by an evolved PET hydrolase from directed evolution. *Angew. Chem. Int. Ed.* **62**, 1–11 (2023).
3. Lu, H. *et al.* Machine learning-aided engineering of hydrolases for PET depolymerization. *Nature* **604**, 662–667 (2022).
4. Cui, Y. *et al.* Computational redesign of a PETase for plastic biodegradation under ambient condition by the GRAPE strategy. *ACS Catal.* **11**, 1340–1350 (2021).
5. Son, H. F. *et al.* Rational protein engineering of thermo-stable PETase from *Ideonella sakaiensis* for highly efficient PET degradation. *ACS Catal.* **9**, 3519–3526 (2019).
6. Sulaiman, S. *et al.* Isolation of a novel cutinase homolog with polyethylene terephthalate-degrading activity from leaf-branch compost by using a metagenomic approach. *Appl Env. Microbiol* **78**, 1556–1562 (2012).
7. Tournier, V. *et al.* An engineered PET depolymerase to break down and recycle plastic bottles. *Nature* **580**, 216–219 (2020).
8. Gewert, B., Plassmann, M. M. & MacLeod, M. Pathways for degradation of plastic polymers floating in the marine environment. *Environ. Sci. Process. Impacts* **17**, 1513–1521 (2015).
9. Satti, S. M., Shah, A. A., Auras, R. & Marsh, T. L. Isolation and characterization of bacteria capable of degrading poly(lactic acid) at ambient temperature. *Polym Degrad Stab* **144**, 392–400 (2017).
10. Auta, H. S., Emenike, C. U. & Fauziah, S. H. Screening of *Bacillus* strains isolated from mangrove ecosystems in Peninsular Malaysia for microplastic degradation. *Env. Pollut* **231**, 1552–1559 (2017).

-
11. Jia, X., Qin, C., Friedberger, T., Guan, Z. & Huang, Z. Efficient and selective degradation of polyethylenes into liquid fuels and waxes under mild conditions. *Sci. Adv.* **2**, e1501591 (2016).
 12. Barth, M. *et al.* A dual enzyme system composed of a polyester hydrolase and a carboxylesterase enhances the biocatalytic degradation of polyethylene terephthalate films. *Biotechnol. J.* **11**, 1082–1087 (2016).
 13. Mrigwani, A., Thakur, B. & Guptasarma, P. Conversion of polyethylene terephthalate into pure terephthalic acid through synergy between a solid-degrading cutinase and a reaction intermediate-hydrolysing carboxylesterase. *Green Chem.* **24**, 6707–6719 (2022).
 14. Knott, B. C. *et al.* Characterization and engineering of a two-enzyme system for plastics depolymerization. *Proc. Natl. Acad. Sci.* **117**, 25476–25485 (2020).
 15. Bell, E. L. *et al.* Directed evolution of an efficient and thermostable PET depolymerase. *Nat Catal* **5**, 673–681 (2022).
 16. Serghei, A., Tress, M. & Kremer, F. The glass transition of thin polymer films in relation to the interfacial dynamics. *J. Chem. Phys.* **131**, 154904 (2009).
 17. Xia, W., Hsu, D. D. & Ketten, S. Molecular Weight Effects on the Glass Transition and Confinement Behavior of Polymer Thin Films. *Macromol. Rapid Commun.* **36**, 1422–1427 (2015).
 18. Vertommen, M. A. M. E., Nierstrasz, V. A., Veer, M. van der & Warmoeskerken, M. M. C. G. Enzymatic surface modification of poly(ethylene terephthalate). *J. Biotechnol.* **120**, 376–386 (2005).
 19. Zhang, J., Wang, X., Gong, J. & Gu, Z. A study on the biodegradability of polyethylene terephthalate fiber and diethylene glycol terephthalate. *J. Appl. Polym. Sci.* **93**, 1089–1096 (2004).
 20. Suye, S., Takahashi, Y., Fujita, S. & Sakakibara, M. Screening and isolation of diethyl p-phthalate utilizing microorganisms. *Seni Gakkaishi* **58**, 416–419 (2002).
 21. Liu, J. *et al.* Biodegradation of diethyl terephthalate and polyethylene terephthalate by a novel identified degrader *Delftia* sp. WL-3 and its proposed metabolic pathway. *Lett. Appl. Microbiol.* **67**, 254–261 (2018).
 22. Pinto, A. V. *et al.* Reaction Mechanism of MHETase, a PET Degrading Enzyme. *ACS Catal.* **11**, 10416–10428 (2021).

Reviewer #3:

General Comments to the Authors: *Li et al. describe in this manuscript the discovery of two enzymes (named ChryBHETase and BsEst) which can hydrolyze the compound BHET, a water-soluble intermediate formed in the degradation of PET (using PETases able to act on the large polymer), which is very similar to MHET for which so-called MHETase have already been described in literature (the only difference between MHET and BHET is the presence of an additional ethylene glycol unit present at the terephthalic acid).*

Technically, the authors have used an entire classical approach (shown in Figure 1, b) to identify these two hydrolases: take environmental microbial samples, tests them for the desired activity, identify the microorganism and then the gene encoding the “BHETase”. Produce them recombinantly, purify and biochemically characterize them followed by improving them using typical methods for protein engineering. In addition, they have predicted the structures of both enzymes using AlphaFold2 and compared them to known enzymes active on BHET and/or MHET. Even if they name their approach “mechanism-guided barrier engineering” and name a method “KnowCoverly” strategy, this does not mean that the concepts are novel at all. Researchers perform mechanism-guided enzyme engineering using computational tools and directed evolution since about two decades. As stated above, the authors have used very common approaches to identify novel enzymes for applications in biocatalysis (not only for plastic degradation) and improved them using state-of-the-art methods.

It was not a true pleasure to read the manuscript as the English is not great and extensive language editing is required. Furthermore, many scientific aspects are vague, not clear or not put into the right context (with respect to literature data/reports) as outlined below. Nevertheless, it is an interesting study which in principle deserves publication after thorough revision, however, the concepts/methods used and achievements made are not outstanding.

We greatly thank Reviewer 3 for the comment “*it is an interesting study which in principle deserves publication after thorough revision~~~~*” and for giving the valuable comments to help us improve the manuscript. We carefully revised the entire manuscript, including proofreading for language and readability improvements. To prove the significance of our work, we included a comparison of seven state-of-the-art PET hydrolyzing enzyme systems at high temperatures (60 °C) that have been previously reported, including FAST-PETase ¹, DuraPETase ², ThermoPETase ³,

DepoPETase⁴, LCC and LCC-ICCG⁵ (Please refer to **Results 2.4** in the revised version and **Comment 3 & 5**).

Comment 1:

In the introduction they claim that although many microbes and enzymes have been described for PET degradation (which is correct), ‘researchers are still mining for the last piece of the puzzle’, the BHETase. This was for sure their motivation, but in fact BHET does not represent a major problem in the enzymatic recycling of PET. As shown in the two publications by Tournier et al. (Nature 2020, Ref. 35) and the Alper team (Nature 2022, Ref. 37), which represent the ‘bench-mark’ processes, BHET is hardly detectable during hydrolysis of PET. Even MHET (which has been reported in a few publications to maybe inhibit the PET-hydrolyzing enzyme) does not present a major problem in large scale processing of PET (so this ‘last piece of puzzle’ is not as relevant as they believe/write in the manuscript). It is stated on page 4, line 71, that in the process developed by Tournier et al. using LCC (note: in that process the highly active and thermostable variant LCC-ICCG, is used not LCC wildtype) “a minimum of 90% conversion” is achieved. In fact, they achieve >98% conversion at 200 g/L, which in turn indicates that BHET (or MHET) are hardly present at all during the process and get rapidly hydrolyzed. The same holds true for the many examples reported by the Alper team in their paper (Ref. 37), where also complete conversion of PET was published (without adding a BHET or MHETase).

Discussed and included. Thanks for bringing these to our attention. After carefully reviewing Alper and Tournier’s work, we find both use the total PET monomers (the sum of TPA and MHET) to indicate their PET degradation performance rather than only TPA. Jiang et al. precisely examined the product composition from PET film degradation by LCC-ICCG, which contained significant amounts of BHET and MHET (See data in **Fig. R1** as below¹⁸). Indeed, after a long degradation period, all BHET can be degraded. As shown in **Fig. S32**, the lower efficiency of PET hydrolase than BHETase can significantly increase reaction time and cost. In addition, the small amount of BHET/MHET still results in a non-homogeneous product, affecting the mechanical properties of the resynthesized PET. As shown in the two-enzyme system described in **Section 2.4**, the BHETase found in this study not only ensures the enzymatic hydrolysis of PET to homogeneous TPA, but also further increases the reaction rate and the final TPA yield compared to the single enzyme system. Also, inhibiting intermediates accumulated in the reaction medium is an important factor limiting the polyester hydrolase efficiency¹⁰. And only a tiny fraction of PET hydrolase was free in the solution during the degradation of PET¹¹. Both will lead to, as suggested

by the reviewer, the accumulation of BHET/MHET during the process as the crystallinity increases, thus inhibiting the hydrolase activity.

Figure R1. The time course study of the PET degradation by LCC-ICCG with and without various binding modules regarding the release of HPLC detectable hydrolysis products ¹⁸.

To clarify the above points, we revised and included the following parts in the revised manuscript:

Page 18-19, line 341-361:

As depicted in **Fig. 5b-h**, coupling Δ BHETase with all seven PET hydrolases in a two-enzyme system resulted in a substantial increase in TPA production compared to using PET hydrolase alone in a single-enzyme system. Interestingly, our study did not observe complete hydrolysis of PET films into homogeneous TPA within 120 h for any single PET hydrolases under the current depolymerization conditions (as shown in **Fig. 5b-h**). Instead, significant amounts of BHET+MHET were detected during degradation in almost all single-enzyme system (**Fig. 5i**). These results differ from previous studies on single FAST-PETase ¹³, and LCC-ICCG ⁵. Unfavorable circumstances and/or the crystallinity of the plastics may be contributing factors to this outcome ^{16,17}. In contrast, the incorporation of Δ BHETases into a single depolymerization system at 60 °C resulted in the complete hydrolysis of PET films to TPA, as observed in DepoPETase- Δ BHETase and FAST-PETase- Δ BHETase (**Fig. 5c-d** and **Fig. S31**). Specifically, when combined with DepoPETase, Δ BsEst demonstrated a 1.8-fold TPA yield increase compared to DepoPETase alone. Similarly, in combination with FAST-PETase, Δ ChryBHETase exhibited a 1.6-fold increase, and Δ BsEst achieved an impressive 2.0-fold increase compared to FAST-PETase alone. These notable enhancements can be attributed to the Δ BHETases' higher affinity for the intermediate product BHET and their ability to maintain the superior turnover efficiency than all state-of-the-art PET hydrolase (**Fig. S32**, TPA conversion rate of Δ BsEst is 31.6 mM/h/mg_{enzyme}, and Δ ChryBHETase is 14.4 mM/h/mg_{enzyme}).

Page S52, line 886-895 (SI):

Fig. S2 The reaction curve of Δ BHETases and other PET hydrolases. (a) Depletion curve of BHET as a substrate. (b) Generation curve of TPA as a terminal product. Reaction condition: 5 mM BHET and 0.5mg/mL Δ BHETases at 60 °C in pH 8.5 buffer for 7 h. Error bars represent the standard deviation of three replicate experiments. TPA conversion rate of $\Delta BsEst$ is 31.6 mM/h/mg_{enzyme}, $\Delta ChryBHETase$ is 14.4 mM/h/mg_{enzyme}, DepoPETase is 10.2 mM/h/mg_{enzyme}, FAST-PETase is 10.46 mM/h/mg_{enzyme}, DuraPETase is 8.56 mM/h/mg_{enzyme}, ThermoPETase is 10.44 mM/h/mg_{enzyme}, LCC is 9.56 mM/h/mg_{enzyme}, and LCC-ICCG is 10.12 mM/h/mg_{enzyme}.

Page 4, line 73:

“Also, the engineered **LCC-ICCG** cutinase showed a minimum of 90% PET depolymerization into monomers over 10 hours by introducing the disulfide bridge ⁵.”

Comment 2:

Abstract and main text: they write that the BHETases show up to 3.5-fold enhanced catalytic efficiency and thermostability: Firstly, what has been improved 3.5-fold? Activity or thermostability or both? That makes no sense as both features can't be put into one number of fold improvement. Secondly, 3.5-fold compared to what? The wild-type enzymes? In addition, this number does not tell us much if we don't know how many grams or moles / liter of BHET are hydrolyzed per hour (or minute). It is important that if a BHETase is used in a PET-hydrolysis process in combination with an efficient PETase (as those mentioned in the Nature papers and in their own study), that the BHETase is as efficient in its turnover as the PETase, otherwise such an enzyme is not useful.

Explained and addressed. We fully agree with the reviewer's comments. First, in the abstract and main text, the "3.5-fold" mentioned refers to the fact that the k_{cat}/K_M of Δ BHETases is 3.5-fold higher than that of wild-type BHETases, maintaining a comparable thermostability. Following the reviewer's advice, we calculated the TPA yield rates for the first 6 h for BHETases and Δ BHETases: 61 μ M/min of *BsEst*, 76

uM/min of $\Delta BsEst$, 14.5 uM/min of *Chry*BHETase and 50.91 uM/min of $\Delta Chry$ BHETase, respectively. The experiments of enzymatic activity towards BHET were supplemented and compared with other reported PET hydrolases (e.g., PETase, FAST-PETase, DuraPETase, DepoPETase, LCC, and LCC-ICCG). As shown in **Fig. S32**, Δ BHETases maintain a superior turnover efficiency towards BHET conversion than all state-of-the-art PET hydrolases. Additionally, Mrigwani et al.¹¹ find that PET hydrolase, like LCC prefers to bind to PET-like hydrophobic materials, implying that only a tiny proportion of PET hydrolase is free in solution during PET biodegradation. Also, the results in **Section 2.4** (see **Comment 3**) confirmed the two-enzyme system with BHETases exhibited a remarkable ability to degrade the PET film than the single-enzyme system effectively. Therefore, we are convinced that the newly discovered BHETase plays a crucial role in the puzzle of PET biodegradation patterns.

To clarify the above points, we revised and included the following parts in the revised manuscript:

Page 2, line 26:

Subsequently, mechanism-guided barrier engineering was employed to yield two robust and thermostable Δ BHETases with up to 3.5-fold enhanced k_{cat}/K_M than wild-type, followed by atomic resolution understanding.

Page 19-20, line 357-365:

These notable enhancements can be attributed to the Δ BHETases' higher affinity for the intermediate product BHET and their ability to maintain the superior turnover efficiency than all state-of-the-art PET hydrolase (**Fig. S32**, TPA conversion rate of $\Delta BsEst$ is 31.6 mM/h/mg_{enzyme}, and $\Delta Chry$ BHETase is 14.4 mM/h/mg_{enzyme}). Intriguingly, even without additional enzymes or components, purified $\Delta BsEst$ and $\Delta Chry$ BHETase (data not shown) generated trace amounts of MHET and TPA from the PET film. This discovery underscores the remarkable capability of Δ BHETase to directly degrade PET, serving as a pivotal catalyst in the process.

Page S52, line 886-895 (SI):

Fig. S3 The reaction curve of Δ BHETases and other PET hydrolases. (a) Depletion curve of BHET as a substrate. (b) Generation curve of TPA as a terminal product. Reaction condition: 5 mM BHET and 0.5mg/mL Δ BHETases at 60 °C in pH 8.5 buffer for 7 h. Error bars represent the standard deviation of three replicate experiments. TPA conversion rate of Δ BsEst is 31.6 mM/h/mg_{enzyme}, Δ ChryBHETase is 14.4 mM/h/mg_{enzyme}, DepoPETase is 10.2 mM/h/mg_{enzyme}, FAST-PETase is 10.46 mM/h/mg_{enzyme}, DuraPETase is 8.56 mM/h/mg_{enzyme}, ThermoPETase is 10.44 mM/h/mg_{enzyme}, LCC is 9.56 mM/h/mg_{enzyme}, and LCC-ICCG is 10.12 mM/h/mg_{enzyme}.

Comment 3:

The same “number problem” is true for the “7.0-fold increase” stated in line 30 of the abstract. 7.0-fold increase compared to the process from Tournier? They convert 200 g/L within 10 h, so 7.0-fold increase would be 1400 g/L in 10 h or 200 g/L within 1.6 hours? I seriously doubt that productivity. To claim in line 31 that their BHETase boosts “state-of-the-art” PETase is not supported by any of their experimental data in comparison to the bench-mark processes (see above). In Figure 3d, substrate concentrations are given as 5 mM BHET (and mM BHET/PET in Figure 5 when the two enzymes were used together), this is orders of magnitudes (!) below the concentrations of PET used in the process by Tournier et al. and the Alper team, which both set the bench mark for efficient PET recycling. When they use minimum 200 g/L, the corresponding concentration of BHET formed is significant (if the authors of this paper are right that BHET represents a problem in the process, which I doubt, see above).

Explained and included. In the original manuscript, the “7.0-fold increase” refers to a two-enzyme system (PETase + Δ BHETase) that produces 7 times more TPA from PET degradation than a single-enzyme system (PETase) at 30 °C. According to Reviewer 3’s comments, we utilized the engineered Δ BHETases (Δ BsEst and Δ ChryBHETase) in a two-enzyme system to successfully generate homogeneous TPA through PET biodegradation. We demonstrated the feasibility of coupling Δ BHETases

with all seven state-of-the-art PET hydrolases, namely PETase¹, DepoPETase⁴, FAST-PETase¹³, DuraPETase², ThermoPETase³, LCC¹⁴ and LCC-ICCG⁵. Detailed results are described **in section 2.4 and Text 9 in SI** (see below). As depicted in **Fig 5c-i**, the coupling of Δ BHETase with all seven PET hydrolases in a two-enzyme system resulted in a substantial increase in TPA production compared to using PET hydrolase alone in a single-enzyme system.

We have modified **Abstract** and **Result 2.4** in the revised manuscript to clarify this point.

Page 2, line 28-32:

Coupling Δ BHETase into a two-enzyme system overcomes heterogeneous product formation challenge and results in up to 7.0-fold improved TPA production than seven state-of-the-art PET hydrolases.

Page 18-22, line 324-393:

2.4 Full TPA conversion from PET achieved by Δ BHETase in a simple two-enzyme degradation system

In 2016, Yoshida et al. published a groundbreaking study introducing *Is*PETase¹. This enzyme exhibited superior activity to Tfh, LCC, and FsC when degrading commercial bottle-derived PET and PET film under ambient conditions. This discovery shed light on the potential to mitigate the environmental impact of uncollectable PET release, offering a promising avenue beyond current recycling methods⁶⁻⁹. Nevertheless, TPA produced by PETase often suffered from contamination by oligo ethylene terephthalates, BHET, and MHET, which posed limitations on downstream applications¹⁰⁻¹². To address this challenge, we have developed a two-enzyme degradation system that combines different PET hydrolases, including PETase¹, DepoPETase⁴, FAST-PETase¹³, DuraPETase², ThermoPETase³, LCC¹⁴ and LCC-ICCG⁵, with Δ BHETase. This innovative system enables the production of pure TPA, as illustrated in **Fig. 5a**.

To investigate the viability of Δ BHETase at high temperatures, we conducted experiments under a representative reaction condition of 60 °C. This choice of temperature aligns with commonly employed conditions in various PET degradation studies^{4,5,13,15}. As depicted in **Fig. 5b-h**, coupling Δ BHETase with all seven PET hydrolases in a two-enzyme system resulted in a substantial increase in TPA production compared to using PET hydrolase alone in a single-enzyme system. Interestingly, our study did not observe complete hydrolysis of PET films into homogeneous TPA within

120 h for any single PET hydrolases under the current depolymerization conditions (as shown in **Fig. 5b-h**). Instead, significant amounts of BHET+MHET were detected during degradation in almost all single-enzyme system (**Fig. 5i**). These results differ from previous studies on single FAST-PETase¹³, and LCC-ICCG⁵. Unfavorable circumstances and/or the crystallinity of the plastics may be contributing factors to this outcome^{16,17}. In contrast, the incorporation of Δ BHETases into a single depolymerization system at 60 °C resulted in the complete hydrolysis of PET films to TPA, as observed in DepoPETase- Δ BHETase and FAST-PETase- Δ BHETase (**Fig. 5c-d** and **Fig. S31**). Specifically, when combined with DepoPETase, Δ BsEst demonstrated a 1.8-fold TPA yield increase compared to DepoPETase alone. Similarly, in combination with FAST-PETase, Δ ChryBHETase exhibited a 1.6-fold increase, and Δ BsEst achieved an impressive 2.0-fold increase compared to FAST-PETase alone. These notable enhancements can be attributed to the Δ BHETases' higher affinity for the intermediate product BHET and their ability to maintain the superior turnover efficiency than all state-of-the-art PET hydrolase (**Fig. S32**, TPA conversion rate of Δ BsEst is 31.6 mM/h/mg_{enzyme}, and Δ ChryBHETase is 14.4 mM/h/mg_{enzyme}). Intriguingly, even without additional enzymes or components, purified Δ BsEst and Δ ChryBHETase (data not shown) generated trace amounts of MHET and TPA from the PET film. This discovery underscores the remarkable capability of Δ BHETase to directly degrade PET, serving as a pivotal catalyst in the process.

The discovery mentioned above was further supported by the analysis of the PET film surface using visible photography, scanning electron microscopy (SEM), and the measurement of reduced water contact angle (**Fig. 5j**). Specifically, the two-enzyme system exhibited a remarkable ability to effectively degrade the PET film with up to 22° water contact angle decrease, resulting in a visibly rougher surface compared to the single enzyme system. To address the additional energy costs associated with high temperatures, we also investigated the PET degradation efficiency of the two-enzyme system at room temperature (30 °C) to achieve a more environmentally friendly process. And we observed similar achievements at room temperature compared to at 60 °C (**Fig. S33**). Remarkably, PETase/ Δ BsEst and PETase/ Δ ChryBHETase produced 663.1 μ M (7.0-fold) and 617.6 μ M (6.5-fold) of TPA within 24 h, far more than PETase alone (95 μ M). Summarily, these significant achievements reinforce our conviction that the newly discovered BHETase plays a crucial role in the puzzle of PET biodegradation patterns. Further details regarding the two-enzyme system can be found in **Text 9** and **Fig. S33** in **SI**.

Fig. 5 Two-enzyme degradation system at 60 °C. (a) The overview of the two-enzyme degradation system. Time course of PET depolymerization in a two-enzyme system with different PET hydrolases, including (b) PETase, (c) DepoPETase, (d) FAST-PETase, (e) DuraPETase, (f) ThermoPETase, (g) LCC, and (h) LCC-ICCG. (i)

The PET monomers (the sum of BHET and MHET) released at 96 h. (j) The SEM images (up panel), and water contact angle analysis (down panel) of the PET film in a single-enzyme degradation system, two-enzyme degradation system with $\Delta BsEst$, and two-enzyme degradation system with $\Delta ChryBHETase$ after 48 h at 60 °C. Reaction condition: The PET films ($\phi=6$ mm) were soaked in 2940 μ L of Na_2HPO_4 - NaH_2PO_4 (pH 8.5, 50 mM) buffer at 60 °C with 50 μ L of 0.5 mg/mL PET hydrolases and $\Delta BHETases$, respectively. The fresh enzyme solution was supplemented every 24 h for depolymerization. Error bars represent the standard deviation of three replicate experiments.

Comment 4:

For a better comparison with the benchmark PETases, they should better report the activity of the BHETase as gram TPA formed per liter and hour and provide the specific productivity (e.g., TPA formed per liter and hour and g enzyme). Consequently, phrases like “the excellent BHET degradation performance...” are wrong and misleading.

Addressed. We fully agree with the reviewer’s comments. We calculated the specific productivity for each enzyme with BHET as the substrate. $\Delta BsEst$ has a particular productivity of 31.6 mM/h/mg enzyme, $\Delta ChryBHETase$ has a specific productivity of 14.4 mM/h/mg enzyme, and PETase has a particular productivity of 0.92 mM/h/mg enzyme. And the rest PET hydrolases all showed lower TPA generation efficiency than $\Delta BsEst$ and $\Delta ChryBHETase$.

To clarify the above points, we have revised and emphasized the following sentences in the revised manuscript:

Page 19, line 357-361:

These notable enhancements can be attributed to the $\Delta BHETases$ ’ higher affinity for the intermediate product BHET and their ability to maintain the superior turnover efficiency than all state-of-the-art PET hydrolase (Fig. S32, TPA conversion rate of $\Delta BsEst$ is 31.6 mM/h/mg_{enzyme}, and $\Delta ChryBHETase$ is 14.4 mM/h/mg_{enzyme}).

Comment 5:

The authors could improve the thermostability of the BHETase, but the best temperature values studied/achieved appear still too low to enable a combination with the PETase designed by Tournier et al. (Ref. 35, process runs at 70°C to overcome the crystallinity of PET) or by the Alper team (Ref. 37, process runs around 50°C). to achieve complete PET hydrolysis.

Explained and addressed. We agree that the data of the two-enzyme system at high temperature (60 °C) should be compared in detail with other state-of-the-art polyester degrading enzymes, such as PETase¹, DepoPETase⁴, FAST-PETase¹³, DuraPETase², ThermoPETase³, LCC¹⁴ and LCC-ICCG⁵). To clarify this point, we modified **Result 2.4** and **Figure 5** in the revised manuscript and **Comment 3**).

Comment 6:

Synthesis of p-Phthaloyl chloride: this is not of much interest and not new. There are many possible chemical reactions to be done with terephthalic acid (TPA) obtainable from PET hydrolysis. To make this halogenated precursor does not fit at all into this manuscript. Moreover, I believe that it would be much better for the climate, if the TPA obtainable from PET hydrolysis is used to make new PET polymer (as demonstrated in Ref. 35 and 37) so that no new petrol must be used to make PET, rather than doing old-fashioned chlorine chemistry.

Addressed and included.

This is an insightful point. As suggested by reviewer 1, we developed a fully closed-loop PET recycling process utilizing Δ BHETases to produce the pure monomer TPA, followed by resynthesizing new PET (**Section 2.5, Fig. 6, Fig. S37**). **We fully agree** that only upcycling the TPA into *p*-phthaloyl chloride cannot meet the PET market demand in society. In **Section 2.5, we successfully resynthesized** the produced TPA monomer obtained from chemical-enzymatic catalysis into virgin PET, achieving up to 554 mg out of 920 mg TPA. The latter accomplished a complete cycle of PET from degradation to repolymerization. Besides showing the upcycling of plastic waste into a single exemplary chemical *p*-phthaloyl chloride, we have emphasized this PET recycling results in the revised manuscript (**More details were transferred to SI**). The latter highlights the potential of our method for contributing to a more sustainable and circular plastic economy.

Page 2, line 33-37:

Finally, we developed a Δ BHETase-joined tandem chemical-enzymatic approach to valorize 21 commercial post-consumed plastics into virgin PET and an exemplary chemical (*p*-phthaloyl chloride) for achieving the closed-loop PET recycling and open-loop PET upcycling.

Page 5, line 91-103:

Furthermore, we utilized the engineered Δ BHETases (Δ BsEst and Δ ChryBHETase) in a two-enzyme system to generate homogeneous TPA through PET biodegradation. We demonstrated the feasibility of coupling Δ BHETases with all seven

state-of-the-art PET hydrolases, namely PETase¹, DepoPETase⁴, FAST-PETase¹³, DuraPETase², ThermoPETase³, LCC¹⁴ and LCC-ICCG⁵. As biotechnological degradation and recycling of plastics are still in the early stages of the learning curve, we further developed a tandem chemical and biological PET degradation system that leverages the advantages of both depolymerization processes to expand the applicability of BHETases fully. To achieve fully closed-loop PET recycling and open-loop PET upcycling, respectively, the obtained pure TPA was used for resynthesizing recycled PET and valorizing TPA into the valuable chemical product (*p*-phthaloyl chloride as an exemplary chemical).

Page 20, line 363-381:

2.5 The closed-loop recycling and open-loop PET upcycling based on the BHETase-joined tandem chemical and biological PET degradation system

Although our results and similar works confirmed that biotechnological degradation has unprecedented potential for PET recycling^{19,20}, enzymatic hydrolysis often has low substrate loads and slower PET degradation compared to certain thermochemical methods²⁰⁻²³. To fully explore the potential of applying $\Delta BsEst$ and $\Delta ChryBHETase$ into practical PET re/upcycling, we developed a tandem chemical-biological approach to leverage the advantages of chemical and biological depolymerization processes. This approach involves a chemical pre-treatment method called glycolysis at the first stage to efficiently degrade PET into low molecular weight hydrolysate BHET within a short time frame while minimizing excessive decomposition^{24,25}. Subsequently, $\Delta BHETases$ contributed to yield pure TPA from the intermediate BHET.

As shown in **Fig. 6a**, K_2CO_3 as a chemical catalyst was selected due to its biocompatibility compared to the most active glycolysis catalysts but toxic Zn salts²⁶. Twenty-one post-consumer plastic products were collected for practical examination, including beverage packaging, food packaging, and household plastic fields (**Table S12**). To evaluate our tandem degradation strategy, a large sample pool of 21 commercial post-consumed plastic waste with crystallinity ranging from 1.54% to 38.73% was used (**Table S12**). Significantly, the BHET yield obtained from PET glycolysis reached an impressive level of over 81.5% (**Fig. S34a and Fig. S35**). Moreover, after undergoing filtration and recrystallization processes, the purity of the BHET achieved was above 95.5% (**Fig. S34a and Fig. S35**). Moving forward, the $\Delta BHETase$ hydrolysis step resulted in the production of TPA with up to 84.1% yield (**Fig. S34b**). These findings demonstrate the efficiency and effectiveness of the

combined PET glycolysis and Δ BHETase hydrolysis processes for various plastic waste degradation and TPA production.

Motivated by the results mentioned above and to complete a closed-loop PET recycling, we resynthesized a total of 554 mg virgin PET directly from the chemical-enzymatic degradation solution (920 mg of TPA) (**Fig. 6a** and **Fig.S36**). And the entire cycle from degradation to repolymerization can be completed within a few days (**Fig. 6b** and **S37a**). These results confirmed the viability of producing virgin PET from non-petroleum sources using our tandem PET recycling system in a closed-loop manner. Besides, converting plastic waste into various high-value chemicals (e.g., fuels, waxes, lubricants) offers an open-loop upcycling perspective to address the urgent issues of plastic waste. Herein, *p*-phthaloyl chloride, as a single exemplary chemical, was synthesized using the TPA yielding from plastic wastes to achieve an open-loop PET upcycling (**Fig. 6c** and **Fig. S37b**, see more details in **SI Text 10**). Collectively, leveraging the engineered Δ BHETase, we have successfully demonstrated a process concept that BHETase-joined tandem PET degradation system to transform plastic waste into valuable products. These innovative technologies hold tremendous potential in supporting a circular plastics economy, where plastic waste is efficiently recycled and repurposed.

Fig. 6 | The closed-loop recycling and open-loop PET upcycling based on the BHETase-joined tandem chemical and biological PET degradation system. (a) The schematic of the closed-loop PET recycling and open-loop PET upcycling pathway. (b) The schematic of the closed-loop PET recycling. (c) The schematic of the open-loop PET upcycling. (d) Theoretical maximum synthesis yield of *p*-phthaloyl chloride from 21 commercial pc-products. Chemical glycolysis conditions: glycolysis of 5 g PET, 3 h of reaction time, 200 °C of temperature, and 0.1 g of K_2CO_3 loading. Enzymatic catalysis condition: 2ml of 0.5 mg/ml $\Delta BsEst$ and $\Delta Chry$ BHETase hydrolyzing the BHET obtained from PET glycolysis at 30 °C within 24 h, respectively. Error bars represent the standard deviation of three replicate experiments.

Fig. S35 Liquid chromatography-mass spectrometry (LC-MS) data of BHET, TPA, and DMT. (a) LC-MS data of BHET by chemical glycolysis; (b) LC-MS data of TPA after enzymatic catalysis by Δ BHETase; (c) LC-MS data of DMT synthesized from TPA. (d) LC-MS data of *p*-phthaloyl chloride synthesized from TPA.

Fig. S36 DSC trace of virgin PET regenerated from the degraded solutions. The crystallinity of this regenerated PET is 45.78%. The melting onset is 210.49 °C. The melting peak temperature is 255.99 °C.

Comment 7:

The enzymes were found in typical microbes known to produce carboxylesterases (especially Bacillus species are known to make esterases very useful for various biocatalytic applications). Thus, it is not unlikely to find an esterase in a Bacillus strain, which can also act on the rather simple water-soluble ester BHET. To call them now BHETase is in principle correct as they used BHET as substrate, found activity and added 'ase' to the compound. Most likely the "BHETase" can act on numerous other esters.

Addressed and included. We fully agree that more tests would help assess the substrate specificity of two BHETases (*BsEst* and *ChryBHETase*). Therefore, we have performed substrate specificity experiments on *pNP* ester substrates of different carbon chain lengths, phenyl esters, glycerol esters and tertiary alcohols, and the relevant content and data have been added to the **Supplementary information**.

Line 205-209 (SI):

Additionally, the substrate specificity of BHETases for other phenyl esters, glycerides, tertiary alcohols were also investigated in **Fig. S7b**. Results indicated that *BsEst* showed significant hydrolysis of almost all substrates tested in this study, while *ChryBHETase* showed significant hydrolysis of 2-methylbutyl acetate, glyceryl tributryate and olive oil substrates.

Line 702-712 (SI):

Fig. S7 The substrate specificity of *BsEst* and *ChryBHETase*. (a) Specific activity of *BsEst* and *ChryBHETase* with different carbon chain lengths of *pNPB* (C₄), *pNPC* (C₈), and *pNPP* (C₁₂) as substrate. Reaction condition: the crude enzymes were incubated with *pNPB*, *pNPC* and *pNPP* in a buffer containing 50 mM Na₂HPO₄-NaH₂PO₄ (pH 7.5) at 40 °C. Error bars represent the standard deviation of three replicate experiments. (b) Substrate specificity of *BsEst* and *ChryBHETase* for other phenyl esters, glycerides, tertiary alcohols. Abbreviation: GTO: Glycerol trioctanoate; PA: Phenyl acetate; 2-MA: 2-Methylbutyl acetate; LA: Linalyl acetate; GTB: Glycerol tributrate; OO: Olive oil. The deeper the yellow, the greater hydrolysis capacity.

Comment 8:

They often misuse the terms “degradation” and “recycling”. Recycling is definitely preferred as the monomer building blocks can be accessed and used to make new polymer. This has been achieved for PET (e.g., by the Tournier et al. and Lu H. et al. teams, Ref. 35 and 37). Degradation does not imply that the monomers can be accessed to make new plastics, also this is impossible for e.g. polyethylene or PVC, where degradation can only yield low molecular weight compounds, but for sure neither ethylene nor vinylchloride as the double bonds reacted in the polymerization). Thus, the use of both terms “degradation” and “recycling” should be carefully checked in future manuscript versions. When using microbes (instead of isolated enzymes), even for PET recycling is unlikely as the products released can be metabolized by the microbes, which is rather a downside.

Checked and revised. We fully agree with your comments and have carefully reviewed our use of the terms “degradation” and “recycling”. We will be more cautious and accurate in future manuscript versions in using these terms.

Reference

1. Yoshida, S. et al. A bacterium that degrades and assimilates poly(ethylene terephthalate). *Science* **351**, 1196–1199 (2016).

-
2. Cui, Y. *et al.* Computational redesign of a PETase for plastic biodegradation under ambient condition by the GRAPE strategy. *ACS Catal.* **11**, 1340–1350 (2021).
 3. Son, H. F. *et al.* Rational protein engineering of thermo-stable PETase from *Ideonella sakaiensis* for highly efficient PET degradation. *ACS Catal.* **9**, 3519–3526 (2019).
 4. Shi, L. *et al.* Complete depolymerization of PET waste by an evolved PET hydrolase from directed evolution. *Angew. Chem. Int. Ed.* **62**, 1–11 (2023).
 5. Tournier, V. *et al.* An engineered PET depolymerase to break down and recycle plastic bottles. *Nature* **580**, 216–219 (2020).
 6. Gewert, B., Plassmann, M. M. & MacLeod, M. Pathways for degradation of plastic polymers floating in the marine environment. *Environ. Sci. Process. Impacts* **17**, 1513–1521 (2015).
 7. Satti, S. M., Shah, A. A., Auras, R. & Marsh, T. L. Isolation and characterization of bacteria capable of degrading poly(lactic acid) at ambient temperature. *Polym Degrad Stab* **144**, 392–400 (2017).
 8. Auta, H. S., Emenike, C. U. & Fauziah, S. H. Screening of Bacillus strains isolated from mangrove ecosystems in Peninsular Malaysia for microplastic degradation. *Env. Pollut* **231**, 1552–1559 (2017).
 9. Jia, X., Qin, C., Friedberger, T., Guan, Z. & Huang, Z. Efficient and selective degradation of polyethylenes into liquid fuels and waxes under mild conditions. *Sci. Adv.* **2**, e1501591 (2016).
 10. Barth, M. *et al.* A dual enzyme system composed of a polyester hydrolase and a carboxylesterase enhances the biocatalytic degradation of polyethylene terephthalate films. *Biotechnol. J.* **11**, 1082–1087 (2016).
 11. Mrigwani, A., Thakur, B. & Guptasarma, P. Conversion of polyethylene terephthalate into pure terephthalic acid through synergy between a solid-degrading cutinase and a reaction intermediate-hydrolysing carboxylesterase. *Green Chem.* **24**, 6707–6719 (2022).
 12. Knott, B. C. *et al.* Characterization and engineering of a two-enzyme system for plastics depolymerization. *Proc. Natl. Acad. Sci.* **117**, 25476–25485 (2020).
 13. Lu, H. *et al.* Machine learning-aided engineering of hydrolases for PET depolymerization. *Nature* **604**, 662–667 (2022).
 14. Sulaiman, S. *et al.* Isolation of a novel cutinase homolog with polyethylene terephthalate-degrading activity from leaf-branch compost by using a metagenomic approach. *Appl Env. Microbiol* **78**, 1556–1562 (2012).
 15. Bell, E. L. *et al.* Directed evolution of an efficient and thermostable PET depolymerase. *Nat Catal* **5**, 673–681 (2022).
 16. Serghei, A., Tress, M. & Kremer, F. The glass transition of thin polymer films in relation to the interfacial dynamics. *J. Chem. Phys.* **131**, 154904 (2009).
 17. Xia, W., Hsu, D. D. & Keten, S. Molecular Weight Effects on the Glass Transition and Confinement Behavior of Polymer Thin Films. *Macromol. Rapid Commun.* **36**, 1422–1427 (2015).

-
18. Xue, R. *et al.* Fusion of Chitin-Binding Domain From Chitinolytic bacter *meiyuanensis* SYBC-H1 to the Leaf-Branch Compost Cutinase for Enhanced PET Hydrolysis. *Front Bioeng Biotech* **9**, (2021).
 19. Wei, R. *et al.* Possibilities and limitations of biotechnological plastic degradation and recycling. *Nat. Catal.* **3**, 867–871 (2020).
 20. Kim, H. T. *et al.* Chemo-Biological Upcycling of Poly(ethylene terephthalate) to Multifunctional Coating Materials. *ChemSusChem.* **14**, 4251–4259 (2021).
 21. Palm, G. J. *et al.* Structure of the plastic-degrading *Ideonella sakaiensis* MHETase bound to a substrate. *Nat Commun* **10**, 1717 (2019).
 22. Knott, B. C. *et al.* Characterization and engineering of a two-enzyme system for plastics depolymerization. *Proc. Natl. Acad. Sci.* **117**, 25476–25485 (2020).
 23. Barragán-Iglesias, P. *et al.* Inhibition of Poly(A)-binding protein with a synthetic RNA mimic reduces pain sensitization in mice. *Nat. Commun.* **9**, 10 (2018).
 24. Castro, A. M. de & Carniel, A. A novel process for poly(ethylene terephthalate) depolymerization via enzyme-catalyzed glycolysis. *Biochem. Eng. J.* **124**, 64–68 (2017).
 25. de Castro, A. M., Carniel, A., Nicomedes Junior, J., da Conceição Gomes, A. & Valoni, É. Screening of commercial enzymes for poly(ethylene terephthalate) (PET) hydrolysis and synergy studies on different substrate sources. *J. Ind. Microbiol. Biotechnol.* **44**, 835–844 (2017).
 26. López-Fonseca, R., Duque-Ingunza, I., de Rivas, B., Arnaiz, S. & Gutiérrez-Ortiz, J. I. Chemical recycling of post-consumer PET wastes by glycolysis in the presence of metal salts. *Polym Degrad Stab* **95**, 1022–1028 (2010).

REVIEWERS' COMMENTS

Reviewer #1 (Remarks to the Author):

The authors have substantially improved upon their manuscript and adequately addressed all prior concerns that were raised.

Reviewer #2 (Remarks to the Author):

The authors have significantly improved the manuscript in a comprehensive revision process and it is thus now acceptable for publication.

Reviewer #3 (Remarks to the Author):

Li et al. describe in this revised manuscript the discovery of two enzymes (named ChryBHETase and BsEst) which can hydrolyze the compound BHET, a water-soluble intermediate formed in the degradation of PET. Then they made improved variants and finally combined this with PETases in a two-enzyme reaction.

I appreciate the efforts made by the authors to address the comments raised by myself and the other reviewers, which clearly helped to improve the manuscript.

It is clear that with the PET material used by them and under the conditions used in this experimental setup, the (improved) delta-BHETase enzymes help to hydrolyze the PET better in combination with the seven PETases (described in literature in the past few years) as shown in Figures 5b-h. On the other hand, it is surprising that the conversions (TPA released) do not match at all what has been reported e.g., in the publications by Tournier et al. (Nature, 2020) using LCC-ICCG or by Lu et al. (Nature 2022) using the FAST-PETase. They state that “Unfavorable circumstances and/or the crystallinity of plastics could be the root causes”, which I do not find really convincing. When Tournier et al. or Lu et al. report >90% conversion of PET into TPA/ethylene glycol within only 10 hrs (Tournier et al.), how can they get only low conversions with the same enzymes? I wrote in my previous evaluation that Tournier et al. “achieved >98% conversion at 200 g/L”, which was not convincingly discussed/addressed in the revision.

Just based on their data, it is clear that the delta-BHETase helps to convert PET completely, but this does not reflect what the earlier Nature publications have reported, where definitely neither BHET nor MHET represented difficult by-products/intermediates and hence no delta-BHETase (or an MHETase) are needed at all.

In addition, they still claim that their concept of “mechanism-guided barrier engineering” and the “KnowCover” strategy are something new to identify and improve enzymes. This is a very classical approach (take environmental microbial samples, tests them for the desired activity, identify the microorganism and then the gene encoding the enzyme of interest, express it recombinantly, purify and characterize it and then use typical methods of protein engineering).

The English language has not been much improved. The paper still contains a range of sentences and phrasings with poor grammar and wordings. A native speaker must carefully improve the paper before acceptance should be considered.

Altogether, this is an interesting study, experimentally overall well performed and nicely illustrated. It for sure deserves publication after another round of revision to address the points given above, but I am not convinced that the concepts/methods used and achievements made are outstanding enough for a publication in a highest-ranking journal.

The original comments of the reviewers are in italics.

Manuscript Number: NCOMMS-23-02054A-Z

Manuscript Title: Discovery and mechanism-guided engineering of BHET hydrolases for improved PET recycling and upcycling

Reviewer #1 and #2:

General Comments to the Authors: *The authors have substantially improved upon their manuscript and adequately addressed all prior concerns that were raised.*

The authors have significantly improved the manuscript in a comprehensive revision process and it is thus now acceptable for publication.

We greatly thank Reviewer #1 and #2 for considering our revised manuscript for publication in *Nature Communications*.

Reviewer #3:

General Comments to the Authors: *Li et al. describe in this revised manuscript the discovery of two enzymes (named ChryBHETase and BsEst) which can hydrolyze the compound BHET, a water-soluble intermediate formed in the degradation of PET. Then they made improved variants and finally combined this with PETases in a two-enzyme reaction.*

I appreciate the efforts made by the authors to address the comments raised by myself and the other reviewers, which clearly helped to improve the manuscript. It is clear that with the PET material used by them and under the conditions used in this experimental setup, the (improved) delta-BHETase enzymes help to hydrolyze the PET better in combination with the seven PETases (described in literature in the past few years) as shown in Figures 5b-h. On the other hand, it is surprising that the conversions (TPA released) do not match at all what has been reported e.g., in the publications by Tournier et al. (Nature, 2020) using LCC-ICCG or by Lu et al. (Nature 2022) using the FAST-PETase. They state that “Unfavorable circumstances and/or the crystallinity of plastics could be the root causes”, which I do not find really convincing. When Tournier et al. or Lu et al. report >90% conversion of PET into TPA/ethylene glycol within only 10 hrs (Tournier et al.), how can they get only low conversions with the same enzymes? I wrote in my previous evaluation that Tournier et al. “achieved >98% conversion at 200 g/L”, which was not convincingly discussed/addressed in the revision. Just based on their data, it is clear that the delta-BHETase helps to convert PET completely, but this does not reflect what the earlier Nature publications have reported, where definitely neither BHET nor MHET represented difficult by-products/intermediates and hence no delta-BHETase (or an MHETase) are needed at all.

In addition, they still claim that their concept of “mechanism-guided barrier engineering” and the “KnowCoverly” strategy are something new to identify and improve enzymes. This is a very classical approach (take environmental microbial samples, tests them for the desired activity, identify the microorganism and then the gene encoding the enzyme of interest, express it recombinantly, purify and characterize it and then use typical methods of protein engineering).

The English language has not been much improved. The paper still contains a range of sentences and phrasings with poor grammar and wordings. A native speaker must carefully improve the paper before acceptance should be considered.

Altogether, this is an interesting study, experimentally overall well performed and nicely illustrated. It for sure deserves publication after another round of revision to address the points given above, but I am not convinced that the concepts/methods used and achievements made are outstanding enough for a publication in a highest-ranking journal.

We greatly thank you for commenting that “*Altogether, this is an interesting study, experimentally overall well performed and nicely illustrated*”. According to the Reviewer #3’s comments, we removed the claims of methodological novelty by replacing the “KnowCovery” strategy to “an enzyme mining process for identifying BHETase (BHETase mining process)” in the main text and supplementary information. Additionally, we did the proofreading of the main text for language and readability with the help of the native speaker (The modified parts are colored in blue).

Explained and discussed.

In our study, we added 0.37 milligrams per gram of PET to the PET degradation system, which was a similar enzyme amount as Lu et al. (0.368 milligrams per gram of PET after Molar unit conversions ¹). And the reaction was performed at 60 °C, which is higher than the optimum temperature of FAST-PETase (50 °C ¹). Tournier et al. added an enzyme concentration of 3 milligrams per gram of PET ², which is 8.1 times enzyme using than ours. We have added detailed reaction condition in the figure caption to clarify the difference with the studies by Tournier et al. and Lu et al ².

Besides, under our reaction condition, we did observe significant BHET+MHET residual even after 96 hours (Fig. 5i), and similar results were obtained by Jiang et al.³ with using LCC-ICCG (“*In all samples, mono (2-hydroxyethyl) terephthalate (MHET) is the most abundant product followed by terephthalic acid (TPA), and bis (2-hydroxyethyl) terephthalate (BHET). During the total incubation time up to 12 h, the levels of released TPA and MHET increased continuously. After 12 h reaction at 65°C, LCC-ICCG yielded 2.12 mM hydrolysis products composed of 0.76 mM TPA, 1.25 mM MHET and 0.11 mM BHET (Figure 4A).*”). Therefore, unfavorable circumstances (e.g., physical-chemical properties of PET) and/or the reaction condition (e.g., enzyme amount and purity, buffer type, pH) may be contributing factors to this outcome. Since our work is aim to overcome the incomplete degradation of PET and low efficiency, we are convinced that the discovered BHETase can be served as a pivotal member in multiple PET biodegradation processes.

To address the above issues, we revised the following sentences in the main text.

Line 348, page 19:

Instead, significant amounts of BHET and MHET were detected during degradation in almost all single-enzyme systems at 96 h (Fig. 5i). Similar results were observed by Jiang et al. with using LCC-ICCG³, but slightly differ from previous studies on single FAST-PETase¹, and LCC-ICCG². The enzymatic reaction conditions (e.g., enzyme amount and purity, buffer type, pH) and the physical-chemical properties of specific PET (e.g., crystallization, absorption, scalping, intrinsic viscosity⁴⁻⁷) may be contributing factors to this outcome.

Line 356, page 20:

This discovery underscores the remarkable capability of Δ BHETase to serve as a pivotal catalyst in the PET degradation process.

Line 383, page 22:

Reaction condition: The PET films ($\phi=6$ mm, roughly 67 mg) were soaked in 2940 μ L of Na₂HPO₄-NaH₂PO₄ (pH 8.5, 50 mM) buffer at 60 °C with 50 μ L of 0.5 mg/mL PET hydrolases and Δ BHETases, respectively. The fresh enzyme solution was supplemented every 24 hours for depolymerization.

Reference

1. Lu, H. *et al.* Machine learning-aided engineering of hydrolases for PET depolymerization. *Nature* **604**, 662–667 (2022).
2. Tournier, V. *et al.* An engineered PET depolymerase to break down and recycle plastic bottles. *Nature* **580**, 216–219 (2020).
3. Xue, R. *et al.* Fusion of Chitin-Binding Domain From Chitinolytic bacter *meiyuanensis* SYBC-H1 to the Leaf-Branch Compost Cutinase for Enhanced PET Hydrolysis. *Front. Bioeng. Biotech.* **9**, (2021).
4. Serghei, A., Tress, M. & Kremer, F. The glass transition of thin polymer films in relation to the interfacial dynamics. *The Journal of Chemical Physics* **131**, 154904 (2009).
5. Xia, W., Hsu, D. D. & Keten, S. Molecular Weight Effects on the Glass Transition and Confinement Behavior of Polymer Thin Films. *Macromolecular Rapid Communications* **36**, 1422–1427 (2015).
6. Zimmermann, L., Dombrowski, A., Völker, C. & Wagner, M. Are bioplastics and plant-based materials safer than conventional plastics? In vitro toxicity and chemical composition. *Environ. Int.* **145**, 106066 (2020).
7. Perera, S., Arulrajah, A., Wong, Y., Maghool, F. & Horpibulsuk, S. Evaluation of shear strength properties of unbound PET plastic in blends with demolition wastes. *Constr. Build. Mater.* **262**, 120545 (2020).